# Private Algorithms for Stochastic Saddle Points and Variational Inequalities: Beyond Euclidean Geometry

**Raef Bassily**
Department of Computer Science & Engineering
Translational Data Analytics Institute (TDAI)
The Ohio State University
bassily.1@osu.edu

**Cristóbal Guzmán**
Inst. for Mathematical and Comput. Eng.
Fac. de Matemáticas and Esc. de Ingeniería
Pontificia Universidad Católica de Chile
crguzmanp@uc.cl

**Michael Menart**
Department of Computer Science & Engineering
The Ohio State University *
Department of Computer Science, University of Toronto
Vector Institute
menart.2@osu.edu

## Abstract

In this work, we conduct a systematic study of stochastic saddle point problems (SSP) and stochastic variational inequalities (SVI) under the constraint of $(\epsilon, \delta)$-differential privacy (DP) in both Euclidean and non-Euclidean setups. We first consider Lipschitz convex-concave SSPs in the $\ell_p/\ell_q$ setup, $p, q \in [1, 2]$. That is, we consider the case where the primal problem has an $\ell_p$-setup (i.e., the primal parameter is constrained to an $\ell_p$ bounded domain and the loss is $\ell_p$-Lipschitz with respect to the primal parameter) and the dual problem has an $\ell_q$ setup. Here, we obtain a bound of $\tilde{O}\left(\frac{1}{\sqrt{n}} + \frac{\sqrt{d}}{n\epsilon}\right)$ on the strong SP-gap, where $n$ is the number of samples and $d$ is the dimension. This rate is nearly optimal for any $p, q \in [1, 2]$. Without additional assumptions, such as smoothness or linearity requirements, prior work under DP has only obtained this rate when $p = q = 2$ (i.e., only in the Euclidean setup). Further, existing algorithms have each only been shown to work for specific settings of $p$ and $q$ and under certain assumptions on the loss and the feasible set, whereas we provide a general algorithm for DP SSPs whenever $p, q \in [1, 2]$. Our result is obtained via a novel analysis of the recursive regularization algorithm. In particular, we develop new tools for analyzing generalization, which may be of independent interest. Next, we turn our attention towards SVIs with a monotone, bounded and Lipschitz operator and consider $\ell_p$-setups, $p \in [1, 2]$. Here, we provide the first analysis which obtains a bound on the strong VI-gap of $\tilde{O}\left(\frac{1}{\sqrt{n}} + \frac{\sqrt{d}}{n\epsilon}\right)$. For $p - 1 = \Omega(1)$, this rate is near optimal due to existing lower bounds. To obtain this result, we develop a modified version of recursive regularization. Our analysis builds on the techniques we develop for SSPs as well as employing additional novel components which handle difficulties arising from adapting the recursive regularization framework to SVIs.

---

*This work was done while M. Menart was at The Ohio State University.

38th Conference on Neural Information Processing Systems (NeurIPS 2024).

# 1 Introduction

Stochastic saddle point problems (SSP), are an increasingly important part of the machine learning toolkit. These problems model optimization settings with an inherent min-max structure, and for this reason are also referred to as stochastic min-max optimization problems. Concretely, the goal is to find an approximate solution of the following problem defined over a convex-concave loss,

$$\min_{w \in \mathcal{W}} \max_{\theta \in \Theta} \left\{ F_{\mathcal{D}}(w, \theta) := \mathbb{E}_{x \sim \mathcal{D}}[f(w, \theta; x)] \right\}, \tag{1}$$

where $\mathcal{D}$ is an unknown distribution for which we have access to an i.i.d. sample $S$. Problems of this kind have important applications in stochastic optimization [NJLS09, JNT11, ZL15], federated learning [MSS19], distributionally robust learning [YLMJ22, ZZZ+24a, ZB24], reinforcement learning [DSL+18], and algorithmic fairness [ABD+18, WM19].

Closely related to saddle point problems are stochastic variational inequalities (SVIs). Given a monotone operator, $G_{\mathcal{D}}(z) := \mathbb{E}_{x \sim \mathcal{D}}[g(z; x)]$, the objective is to approximate the point $z^* \in \mathcal{Z}$, where

$$\langle G_{\mathcal{D}}(z^*), z^* - z \rangle \leq 0, \ \ \forall z \in \mathcal{Z}. \tag{2}$$

Stochastic saddle point problems can be easily related to variational inequalities by observing that the *saddle operator* (an operator closely related to the gradient) of a convex-concave function is monotone. While SSPs and SVIs are closely related, it can be the case that a problem which can be formulated as a monotone SVI is not easily cast as a convex-concave SSP [JN19].

Parallel to the above, the problem of *privacy* has become increasingly important in the big data era. In this regard, the notion of differential privacy has arisen as the premier standard. Stochastic optimization problems are a natural target for privacy concerns due to the fact that they are frequently formulated using a dataset of (potentially sensitive) individual data records. For many such problems, the constraint of differential privacy necessitates fundamentally new rates and techniques, and as such the formal characterization of these problems is an important task.

Thus far, work on differentially private SSPs and SVIs has focused primarily on Euclidean settings. However, a number of important, including many of those referenced at the start of this section, are naturally formulated in other geometries. Prior to this work, the optimal utility rate for DP SSPs was known only in Euclidean and polyhedral settings. For SVIs, the best achievable utility was unknown in any geometry (including Euclidean), at least under canonical utility measures. In this work, we provide the first systematic study of SSPs and SVIs in general geometries. The new analysis tools we develop lead to optimal rates for a number of these important setups.

## 1.1 Contributions

In this work, we provide the first systematic study of stochastic saddle point problems and variational inequalities in both Euclidean and non-Euclidean geometries. Our first results pertain to stochastic saddle point problems where the primal problem has an $\ell_p$-setup and the dual problem has an $\ell_q$-setup, where $p, q \in [1, 2]$. Here we assume the convex-concave loss is Lipschitz. We generalize the recursive regularization framework developed in previous works to more handle non-Euclidean geometries [AZ18, BGM23]. At the heart of this extension is a fundamentally new analysis of the generalization properties of this algorithm. The issue of generalization has in fact been a key issue at the heart of many other works studying SSPs [LYYY21, OPZZ22, BGM23], as the presence of a supremum in the strong SP-gap accuracy measure (see Eqn. (4)) breaks more traditional generalization techniques. In contrast to prior work, our generalization technique works by avoiding entirely any generalization bound for the strong gap itself. Rather, we introduce a new accuracy measure measure which, when used in conjunction with the recursive regularization algorithm, eventually translates into a strong gap guarantee. Our technique stands in particular contrast to [BGM23], which is thus far the only work in the DP literature to obtain optimal strong SP-gap rates in the Euclidean setting and also uses recursive regularization. However, their technique fundamentally relies on a McDiarmids style concentration bound that is worse by a $poly(d)$ factor in non-Euclidean setups such as the $\ell_1$ setting [Pan08]. Using these new techniques, we provide the first analysis which obtains the near optimal rate of $\tilde{O}\left(\frac{1}{\sqrt{n}} + \frac{\sqrt{d}}{n\epsilon}\right)$ on the strong SP-gap for any $p, q \in [1, 2]$. Our algorithm achieves this rate in $\tilde{O}\left(\min\left\{\frac{n^2 \epsilon^{1.5}}{\sqrt{d}}, n^{3/2}\right\}\right)$ number of gradient evaluations. We note that the near optimality of this rate

is established by lower bounds for DP stochastic convex optimization, which is a special case of DP SSPs [BGN21, AFKT21]. Previously, comparable rates on the strong gap had only been obtained in the case where $p = q = 2$ or under strong additional assumptions [BGM23, ZB24, GGP24].

Next, we consider DP stochastic variational inequalities with a monotone, bounded, and Lipschitz operator. We adapt the recursive regularization framework even further and again leverage a novel generalization analysis. Here, we obtain the rate $\tilde{O}\left(\frac{1}{\sqrt{n}} + \frac{\sqrt{d}}{n\epsilon}\right)$ on the strong VI-gap (see Eqn. (5)) in the $\ell_p$-setting, $p \in [1, 2]$. Our algorithm again achieves this rate in $\tilde{O}\left(\min\left\{\frac{n^2\epsilon^{1.5}}{\sqrt{d}}, n^{3/2}\right\}\right)$ number of gradient evaluations. This is the first result to obtain the near optimal convergence rate on the strong VI-gap for $p - 1 = \Omega(1)$, which notably includes the Euclidean case. The corresponding lower bound for $p = 2$ was established in [BG23], and we provide a simple extension of their technique to the case where $p - 1 = \Omega(1)$. See Appendix E. Finally, for the setting $p = 2$, we show that our rate can be achieved in a near linear number of gradient evaluations by leveraging acceleration techniques for Lipschitz and strongly monotone variational inequalities.

## 1.2 Related Work

Differentially private stochastic optimization now has a broad body of work spanning over a decade [JKT12, BST14, JT14, TTZ15, BFTT19, FKT20, AFKT21]. Such work has rigorously characterized the problem of stochastic convex optimization in a variety of geometries. In $\ell_p$ setups, for $p \in [1, 2]$, it is now known that the optimal rate is $\tilde{O}(\frac{1}{\sqrt{n}} + \frac{\sqrt{d}}{n\epsilon})$ for such problems [AFKT21, BGN21]. It has also been shown that additional improvements are possible in the $\ell_1$ setting under smoothness assumptions. The study of stochastic saddle point problems under differential privacy is much less developed, but has nonetheless attracted a surge of recent interest [YHL+22, ZTOH22, BGM23, GGP24]. Without privacy, optimal $O(1/\sqrt{n})$ guarantees on the strong SP-gap have long been known [NY78]. With privacy, the (near) optimal $\tilde{O}\left(\frac{1}{\sqrt{n}} + \frac{\sqrt{d}}{n\epsilon}\right)$ rate was obtained only recently, and then only in the Euclidean setting [BGM23]. In fact, work on DP-SSPs has focused largely on the case where $p = q = 2$, despite the fact that important problems are naturally formulated in other geometries. In particular, the case where $q = 1$ has important applications in distributionally robust optimization, federated learning, and algorithmic fairness. In this regard, the works [GGP24, ZB24] have recently studied DP SSPs with $q = 1$. The work [GGP24] studies the $\ell_1/\ell_1$ setting when the loss is additionally assumed to be smooth and the constraint set is polyhedral; they achieved the rate $\tilde{O}\left(\frac{1}{\sqrt{n}} + \frac{1}{(n\epsilon)^{1/2}}\right)$. We note that smoothness is fundamentally necessary in achieving this dimension independent rate, as otherwise existing lower bounds of $\tilde{\Omega}\left(\frac{1}{\sqrt{n}} + \frac{\sqrt{d}}{n\epsilon}\right)$ hold for such problems [AFKT21]. The work [ZB24] studied the problem of differentially private worst-group risk minimization, which is closely related to DP-SSPs in the $\ell_1/\ell_2$ setting, but requires the loss to have a specific linear structure with respect to the dual parameter. For $\ell_1/\ell_2$ saddle point problems having this structure, their result implies a rate of $O\left(\frac{1}{\sqrt{n}} + \frac{\sqrt{d}}{n\epsilon}\right)$ on the strong gap.

Work on SVIs is less developed. Non-privately, the optimal strong VI-gap rate of $O(\frac{1}{\sqrt{n}})$ was established in [JNT11] (although related techniques trace back to [NY78]). Work on differentially private variational inequalities is limited to the work [BG23]. In the Euclidean setup, this work achieved a rate of $\left(\frac{1}{n^{1/3}} + \frac{\sqrt{d}}{n^{2/3}\epsilon}\right)$ on the strong VI-gap under DP.

## 2 Preliminaries

In this section, we detail preliminaries for stochastic saddle point problems and differential privacy. Both SSPs and SVIs share a similar structure, which we detail first. Throughout, we use $[w, \theta]$ to denote the concatenation of the vectors $w$ and $\theta$. For a function $f$, we let $\nabla f$ denote an arbitrary subgradient selection of $f$. Finally, we let $\mathsf{Unif}(U)$ denote the uniform distribution over the set $U$.

**Stochastic Monotone Operators.** Let $\mathcal{X}$ be some abstract data domain and let $S \sim \mathcal{D}^n$ for $n > 0$ and $\mathcal{D}$ some unknown distribution over $\mathcal{X}$. Let $\|\cdot\|$ be some norm and $\|\cdot\|_*$ its dual. We consider some compact convex parameter space $\mathcal{Z} \subseteq \mathbb{R}^d$ of diameter $B$ with respect to $\|\cdot\|$. Let $\mathcal{B}^d_{\|\cdot\|}(r)$ denote the $d$-dimensional ball of radius $r > 0$ w.r.t. $\|\cdot\|$ centered on zero. We assume there exists

$L > 0$ such that $g : \mathcal{Z} \times \mathcal{X} \mapsto \mathcal{B}^d_{\|\cdot\|_*}(L)$ is a bounded operator and that for any $x \in \mathcal{X}$, $g(\cdot; x)$ is monotone. That is, $\forall z_1, z_2 \in \mathcal{Z}$ it holds that $\langle g(z_1; x) - g(z_2; x), z_1 - z_2 \rangle \geq 0$. We define the empirical and population operators as $G_S(z) = \frac{1}{n} \sum_{i=1}^n g(z; x_i)$ and $G_{\mathcal{D}}(z) = \underset{x \sim \mathcal{D}}{\mathbb{E}} [g(z; x)]$.

**Stochastic Saddle Point Problems.** SSPs have the following structure in addition to the above. Let $d_w, d_\theta \geq 0$ such that $d_w + d_\theta = d$. We assume $\mathcal{Z}$ is the product of the convex compact sets $\mathcal{W} \subseteq \mathbb{R}^{d_w}$ and $\Theta \subseteq \mathbb{R}^{d_\theta}$ equipped with norms $\| \cdot \|_w$ and $\| \cdot \|_\theta$ and having diameters $B_w$ and $B_\theta$ respectively. Then $\mathcal{Z} = \mathcal{W} \times \Theta$. We let $\|[w, \theta]\| = \sqrt{\|w\|_w^2 + \|\theta\|_\theta^2}$, and thus the diameter of $\mathcal{Z}$ satisfies $B \leq \sqrt{B_w^2 + B_\theta^2}$; note the geometric mean of two norms is always a norm.

In SSPs, we consider the case where the monotone operator is the *saddle operator* of a convex-concave loss function $f : \mathcal{W} \times \Theta \times \mathcal{X} \mapsto \mathbb{R}$. The saddle operator is defined as $g(w, \theta; x) = [\nabla_w f(w, \theta; x), -\nabla_\theta f(w, \theta; x)]$ and is always monotone if $f$ is convex-concave. We also define the corresponding population loss and empirical loss functions as $F_{\mathcal{D}}(w, \theta) = \underset{x \sim \mathcal{D}}{\mathbb{E}} [f(w, \theta; x)]$ and $F_S(w, \theta) = \frac{1}{n} \sum_{x \in S} f(w, \theta; x)$ respectively. The boundedness assumption on $g$ means $f$ is $L$-Lipschitz. Concretely, $\forall w_1, w_2 \in \mathcal{W}$ and $\forall \theta_1, \theta_2 \in \Theta$:

$$\text{Lipschitzness:} \qquad |f(w_1, \theta_1; x) - f(w_2, \theta_2; x)| \leq L \|[w_1, \theta_1] - [w_2, \theta_2]\| \qquad (3)$$

Under such assumptions, a solution for problem (1) always exists [Sio58], and is referred to as the *saddle point*. Further, given an SSP (1), we will denote a saddle point as $[w^*, \theta^*]$.

The utility of an approximation to the saddle point is characterized by the strong SP-gap. Given a (randomized) algorithm $\mathcal{A}$ with output $[\mathcal{A}_w(S), \mathcal{A}_\theta(S)] \in \mathcal{W} \times \Theta$, this is defined as

$$\text{Gap}_{\text{SP}}(\mathcal{A}) \quad = \quad \underset{\mathcal{A}, S}{\mathbb{E}} \left[ \max_{\theta \in \Theta} \{F_{\mathcal{D}}(\mathcal{A}_w(S), \theta)\} - \min_{w \in \mathcal{W}} \{F_{\mathcal{D}}(w, \mathcal{A}_\theta(S))\} \right]. \qquad (4)$$

For notational convenience, we define the following function which is closely related to the SP-gap, $\widehat{\text{Gap}}_{\text{SP}}(\bar{w}, \bar{\theta}) = \max_{\theta \in \Theta} \{F_{\mathcal{D}}(\bar{w}, \theta)\} - \min_{w \in \mathcal{W}} \{F_{\mathcal{D}}(w, \bar{\theta})\}$. Usefully, this function is known to be Lipschitz whenever $f$ is Lipschitz.

**Fact 1.** *([BGM23]) If $f$ is $L$-Lipschitz then $\widehat{\text{Gap}}_{\text{SP}}$ is $\sqrt{2}L$-Lipschitz.*

Finally, we define $\ell_p/\ell_q$ saddle point problems as those which, in addition to the above, also have the following structure. Assume $\| \cdot \|_w = \| \cdot \|_p$ and $\| \cdot \|_\theta = \| \cdot \|_q$ and that $\mathcal{W}$ and $\Theta$ have diameters bounded by $B_w$ and $B_\theta$ with respect to $\| \cdot \|_p$ and $\| \cdot \|_q$ respectively. We assume that for any $x \in \mathcal{X}$ that $f(\cdot, \cdot; x)$ is $L_w$ Lipschitz in its first parameter w.r.t. $\| \cdot \|_p$ and $L_\theta$ Lipschitz in its second parameter w.r.t. $\| \cdot \|_q$. Note this implies an overall Lipschitz parameter, as per Eqn. (3), of $L \leq \sqrt{L_w^2 + L_\theta^2}$.

**Stochastic Variational Inequalities.** For SVIs, in addition to the assumption that the monotone operator is $L$-bounded, we will also assume it is $\beta$-Lipschitz. The objective for stochastic variational inequalities is to find an approximation of the (population) equilibrium point $z^*$, where $z^*$ is characterized by Eqn. (2). We refer to such a solution as an equilibrium point. For approximate solutions, the quality of the approximation is characterized by the strong VI-gap:

$$\text{Gap}_{\text{VI}}(\mathcal{A}) \quad = \quad \underset{\mathcal{A}, S}{\mathbb{E}} \left[ \max_{z \in \mathcal{Z}} \{\langle G_{\mathcal{D}}(z), \mathcal{A}(S) - z \rangle\} \right]. \qquad (5)$$

Note that it is not true in general that the VI-gap bounds the SP-gap even when the monotone operator in question is the saddle operator of some convex-concave loss (see Fact 3 in Appendix A). However, in such a case it is true that the equilibrium point of the of the SVI is the saddle point of the corresponding SSP.

We say that an operator $g$ is $\mu$-strongly monotone if for any $z_1, z_2 \in \mathcal{Z}$ that $\langle g(z_1; x) - g(z_2; x), z_1 - z_2 \rangle \geq \frac{\mu}{2} \|z_1 - z_2\|$.

**Stability.** Important to our analysis will be the notion of uniform stability [BE02].

**Definition 1.** *A randomized algorithm $\mathcal{A} : \mathcal{X}^n \mapsto \mathcal{W} \times \Theta$ satisfies $\Delta$-uniform argument stability (UAS) if for any pair of adjacent datasets $S, S' \in \mathcal{X}^n$ it holds that $\underset{\mathcal{A}}{\mathbb{E}} [\|\mathcal{A}(S) - \mathcal{A}(S')\|] \leq \Delta$.*

The notion of strong convexity is often important in analyzing stability. A function $\psi : \mathcal{Z} \mapsto \mathbb{R}$ is $\mu$-strongly convex w.r.t. $\|\cdot\|$ if for any $z, z' \in \mathcal{Z}$ one has $\psi(z) - \psi(z') \geq \langle \nabla \psi(z'), z - z' \rangle + \frac{\mu}{2} \|z - z'\|^2$. We call a function $F : \mathcal{W} \times \Theta \mapsto \mathbb{R}$ $\mu$-strongly-convex/strongly-concave (SC/SC) if for any $\theta \in \Theta$ and $w \in \Theta$ the functions $F(\cdot, \theta)$ and $-F(w, \cdot)$ are $\mu$-strongly-convex.

Notably, adding a SC/SC regularizer leads to stability properties. The following fact results from a more general result for regularized SVIs; see Lemma 5 in Appendix A.

**Lemma 1.** *Let $\psi : \mathcal{W} \times \Theta \mapsto \mathbb{R}$ be $\mu$-strongly-convex/strongly-concave w.r.t. $\|\cdot\|_w$ and $\|\cdot\|_\theta$. Then the algorithm which returns the saddle point of $(w, \theta) \mapsto \frac{1}{n} \sum_{z \in S} f(w, \theta; z) + \psi(w, \theta)$ is $(\frac{2L}{\mu n})$-UAS.*

**Differential Privacy (DP) [DMNS06].** An algorithm $\mathcal{A}$ is $(\epsilon, \delta)$-differentially private if for all datasets $S$ and $S'$ differing in one data point and all events $\mathcal{E}$ in the range of the $\mathcal{A}$, we have, $\mathbb{P}(\mathcal{A}(S) \in \mathcal{E}) \leq e^\epsilon \mathbb{P}(\mathcal{A}(S') \in \mathcal{E}) + \delta$.

## 2.1 Example of Non-Euclidean SSP

One important example of non-Euclidean SSPs arises from the problem of minimizing the worst case risk over multiple populations. This problem has arisen in group distributionally robust optimization to name just one of many applications [SGJ22, ZZZ+24b, NMG24]. Let $f : \mathcal{W} \times \mathcal{X} \mapsto \mathbb{R}$ and $\mathcal{W}$ have a standard $\ell_p$ setup. Consider $k$ distributions, $\mathcal{D}_1, \ldots, \mathcal{D}_k$, and the goal of selecting a model $w \in \mathcal{W}$ which guarantees the lowest worst-case risk for the $k$ distributions above:

$$\min_{w \in \mathcal{W}} \max_{j \in [k]} \mathbb{E}_{x_j \sim \mathcal{D}_j}[f(w; x_j)] = \min_{w \in \mathcal{W}} \max_{\theta \in \Delta} \sum_{j=1}^{k} \theta^{(j)} \mathbb{E}_{x_j \sim \mathcal{D}_j}[f(w; x_j)],$$

where here $\Delta$ denotes the standard $k$-dimensional simplex, and the equality above holds by the maximum principle for convex functions [Bau58]. Given that the feasible set for $\theta$ is a simplex, it is natural to endow this space with the $\ell_1$-geometry. We thus end up with a SSP problem in $\ell_p/\ell_1$ setup.

## 3 A New Analysis for Recursive Regularization

---
**Algorithm 1** Recursive Regularization: $\mathcal{R}_{\mathrm{SSP}}$

---
**Require:** Dataset $S \in \mathcal{X}^n$, loss function $f$, subroutine $\mathcal{A}_{\mathrm{emp}}$, regularization parameter $\lambda \geq \frac{L\kappa}{B\sqrt{n}}$
(where $\|\cdot\|_w^2$ and $\|\cdot\|_\theta^2$ are $\frac{1}{\kappa}$ strongly convex), constraint set diameter $B$, Lipschitz constant $L$.
1: Let $n' = n/\log_2(n)$, and $T = \log_2(\frac{L}{B\lambda})$.
2: Let $S_1, ..., S_T$ be a disjoint partition of $S$ with each $S_t$ of size $n'$ *(which is always possible due to the condition on $\lambda$)*
3: Let $[\bar{w}_0, \bar{\theta}_0]$ be any point in $\mathcal{W} \times \Theta$
4: Define function $(w, \theta, x) \mapsto f^{(1)}(w, \theta; x) = f(w, \theta; x) + 2\lambda \|w - \bar{w}_0\|_w^2 - 2\lambda \|\theta - \bar{\theta}_0\|_\theta^2$
5: **for** $t = 1$ to $T$ **do**
6: $\quad [\bar{w}_t, \bar{\theta}_t] = \mathcal{A}_{\mathrm{emp}}(S_t, f^{(t)}, [\bar{w}_{t-1}, \bar{\theta}_{t-1}], \frac{B}{2^t})$
7: $\quad$ Define $(w, \theta, x) \mapsto f^{(t+1)}(w, \theta; x) = f^{(t)}(w, \theta; x) + 2^{t+1}\lambda \|w - \bar{w}_t\|_w^2 - 2^{t+1}\lambda \|\theta - \bar{\theta}_t\|_\theta^2$
8: **end for**
9: **Output:** $[\bar{w}_T, \bar{\theta}_T]$

---

In this section, we present our modified recursive regularization algorithm, first developed in [AZ18] and extended to Euclidean SSPs in [BGM23]. We then discuss the key components of our analysis needed to obtain our results for non-Euclidean geometries. We conclude the section by applying our general result for recursive regularization to DP $\ell_p/\ell_q$-SSPs.

**Algorithm Overview.** As in [BGM23], our recursive regularization implementation, Algorithm 1, solves a series of regularized saddle point problems defined by $f^{(1)}, ..., f^{(T)}$. The saddle point problem defined in each round of Algorithm 1 is solved using some empirical subroutine, $\mathcal{A}_{emp}$. This subroutine takes as input a subset of the dataset, $S_t$, the regularized loss function for that round, $f^{(t)}$, a starting point, $[\bar{w}_{t-1}, \bar{\theta}_{t-1}]$, and an upper bound on the expected distance to the empirical saddle

point of the problem defined by $S_t$ and $f^{(t)}$. The exact implementation of $\mathcal{A}_{emp}$, Algorithm 3, will be discussed in the next section. Here, we focus on the guarantees of Recursive Regularization given that $\mathcal{A}_{emp}$ satisfies a certain accuracy condition. At each round, the empirical subroutine $\mathcal{A}_{\mathsf{emp}}$ is required to find a point, $[\bar{w}_t, \bar{\theta}_t]$, which is close (under $\|\cdot\|$) to the *empirical* saddle point. Because the scale of regularization doubles each round, this task becomes easier each round. Specifically, [BGM23] observed that implementations of $\mathcal{A}_{\mathsf{emp}}$ which satisfy the notion of relative accuracy succeed at finding such points.

**Definition 2** ($\hat{\alpha}$-relative accuracy). *Given a dataset $S' \in \mathcal{X}^{n'}$, loss function $f'$, and an initial point $[w', \theta']$, we say that $\mathcal{A}_{\mathsf{emp}}$ satisfies $\hat{\alpha}$-relative accuracy w.r.t. the empirical saddle point $[w^*_{S'}, \theta^*_{S'}]$ of $F'_{S'}(w, \theta) = \frac{1}{n'} \sum_{x \in S'} f'(w, \theta; x)$ if, $\forall \hat{D} > 0$, whenever $\mathbb{E}\left[\|[w', \theta'] - [w^*_{S'}, \theta^*_{S'}]\|\right] \leq \hat{D}$, the output $[\bar{w}, \bar{\theta}]$ of $\mathcal{A}_{\mathsf{emp}}$ satisfies $\mathbb{E}\left[F'_{S'}(\bar{w}, \theta^*_{S'}) - F'_{S'}(w^*_{S'}, \bar{\theta})\right] \leq \hat{D}\hat{\alpha}$.*

In contrast to [BGM23], our algorithm uses more general regularization to ensure strong-convexity/strong-concavity with respect to the appropriate norm.

**Guarantees of Recursive Regularization.** Our general result for recursive regularization is stated as follows, and its full proof is given in Appendix B.2.

**Theorem 1.** *Let $\mathcal{A}_{\mathsf{emp}}$ satisfy $\hat{\alpha}$-relative accuracy for any $(5L)$-Lipschitz loss function and dataset of size $n' = \frac{n}{\log(n)}$ and assume $\|\cdot\|_w^2$ and $\|\cdot\|_\theta^2$ are $\frac{1}{\kappa}$-strongly convex under $\|\cdot\|_w$ and $\|\cdot\|_\theta$ respectively. Then Algorithm 1, run with $\mathcal{A}_{\mathsf{emp}}$ as a subroutine and $\lambda = \frac{48}{B}\left(\hat{\alpha}\kappa^2 + \frac{L\kappa^{3/2}}{\sqrt{n'}}\right)$, satisfies*

$$\mathrm{Gap}_{\mathsf{SP}}(\mathcal{R}_{SSP}) = O\left(B\hat{\alpha}\kappa^2 \log(n) + \frac{BL\kappa^{3/2}\log^{3/2}(n)}{\sqrt{n}}\right).$$

The similarity of this result to [BGM23, Theorem 5] and our exposition thus far belies the difficulty of adapting their result to non-Euclidean setups. The key challenge addressed by the analysis of [BGM23] was that of *generalization*. In this regard, their key insight was to use McDiarmid style concentration bounds to show that the *empirical* saddle point obtains non-trivial guarantees on the strong gap. However, such concentration results critically rely on the fact that the underlying norm is Euclidean. A generalization of this concentration to, for example, the $\ell_1$ setup, necessarily incurs an additional $\sqrt{d}$ factor [Pan08]. Thus, a fundamentally new analysis is needed. One should also note that the squared norms $\|\cdot\|_w^2$ may *not* be strongly convex for certain norms, such as $\|\cdot\|_1$. Regardless, in some such cases, we can still leverage this result by modifying the underlying problem, as we will show in Section 4.

**Key Proof Ideas.** We circumvent the above issues by providing a fundamentally new generalization analysis for the intermediate iterates of recursive regularization. Specifically, our analysis avoids entirely any analysis of the strong gap at intermediate stages of the algorithm. Instead, we introduce two new functions, which are similar in nature to the quantity used in the definition of relative accuracy, but are taken with respect to the *population* saddle point. For $t \in [T]$, define $F_{\mathcal{D}}^{(t)}(w, \theta; x) := \mathbb{E}_{x \sim \mathcal{D}}\left[f^{(t)}(w, \theta; x)\right]$ and let $[w_t^*, \theta_t^*]$ be its saddle point; define $F_S^{(t)} := \frac{1}{n'} \sum_{x \in S_t} f^{(t)}(w, \theta; x)$. We are interested in the functions,

$$H_{\mathcal{D}}^{(t)}([w, \theta]) := F_{\mathcal{D}}^{(t)}(w, \theta_t^*) - F_{\mathcal{D}}^{(t)}(w_t^*, \theta) \quad \text{and} \quad H_S^{(t)}(w, \theta) := F_S^{(t)}(w, \theta_t^*) - F_S^{(t)}(w_t^*, \theta). \quad (6)$$

Notably, the strong-convexity/strong-concavity of $F_{\mathcal{D}}^{(t)}$ means that a bound on $H_{\mathcal{D}}^{(t)}([w, \theta])$ yields a bound on $\|[w, \theta] - [w_t^*, \theta_t^*]\|$. Ultimately, finding a point sufficiently close to $[w_t^*, \theta_t^*]$ at each round is all recursive regularization needs to succeed. The question then, is how to obtain guarantees on $H_{\mathcal{D}}^{(t)}$. We accomplish this via a stability-implies-generalization argument.

**Lemma 2.** *Let $f : \mathcal{Z} \times \mathcal{X} \mapsto \mathbb{R}$ be $L$-Lipschitz. Let $[w^*, \theta^*] \in \mathcal{Z}$ be the population saddle point. For any $x \in \mathcal{X}$ define $h([w, \theta]; x) = f(w, \theta^*; x) - f(w^*, \theta; x)$. For $S \sim \mathcal{D}^n$, let $H_S(z) = \frac{1}{n} \sum_{x \in S} h(z; x)$ and $H_{\mathcal{D}}(z) = \mathbb{E}_{x \sim \mathcal{D}}[h(z; x)]$. Then for any $\Delta$-UAS algorithm, $\mathcal{A}$, one has $\mathbb{E}_{S, \mathcal{A}}[H_{\mathcal{D}}(\mathcal{A}(S)) - H_S(\mathcal{A}(S))] \leq 2\Delta L$.*

The proof relies on two main observations. First, it is easy to see that because $f$ is Lipschitz, then $h$ is also Lipschitz. Then, because $H_{\mathcal{D}}$ and $H_S$ can be written as the expectation of $f$ w.r.t. $z \sim \mathcal{D}$ and $z \sim \mathsf{Unif}(S)$ respectively, we can apply standard stability-implies-generalization results to $h$ to obtain the claimed result [BE02]. We provide a full proof in Appendix B.1. This analysis bypasses difficulties of working directly with $\widehat{\mathsf{Gap}}_{\mathsf{SP}}$ experienced by [OPZZ22, BGM23] and other works since, in general, there is *no* function $h$ such that $\widehat{\mathsf{Gap}}_{\mathsf{SP}}(z) = \mathbb{E}_{x \sim \mathcal{D}}[h(z;x)]$. Note also it is important that we have defined $h(z;x)$ w.r.t. to the data independent point $[w_t^*, \theta_t^*]$. Were $[w_t^*, \theta_t^*]$ to depend on $S$, this result would not hold.

Because $H_S^{(t)}$ uses the *population* saddle point in its definition, it may not be immediately clear how one could first minimize $H_S^{(t)}$. Direct access to $H_S^{(t)}$ is in fact not possible without knowledge of $[w_t^*, \theta_t^*]$, which the algorithm does not have. In this regard, we observe that algorithms which minimize the *empirical* gap are powerful enough to minimize $H_S^{(t)}$, even without knowledge of $[w_t^*, \theta_t^*]$, since

$$H_S^{(t)}(w, \theta) = F_S^{(t)}(w, \theta_t^*) - F_S^{(t)}(w_t^*, \theta) \leq \max_{w', \theta'} \left\{ F_S^{(t)}(w, \theta') - F_S^{(t)}(w', \theta) \right\}.$$

Particular to our analysis, we will leverage the fact that the exact empirical saddle point is $(\frac{L}{\lambda n})$-stable and has an empirical gap of 0.

## 4    Optimal Rates for Private $\ell_p/\ell_q$ Saddle Point Problems

**Setup.**    In this section, we apply Theorem 1 to obtain results for $\ell_p/\ell_q$ saddle point problems. In order to do this, we will apply recursive regularization using norms slightly different than the $\ell_p$ and $\ell_q$ norms. Specifically, to solve an $\ell_p/\ell_q$ SSP, we define $\bar{p} = \max\left\{p, 1 + \frac{1}{\log(d)}\right\}$ and $\bar{q} = \max\left\{q, 1 + \frac{1}{\log(d)}\right\}$ and will apply recursive regularization with $\|\cdot\|_w = \frac{1}{B_w}\|\cdot\|_{\bar{p}}$ and $\|\cdot\|_\theta = \frac{1}{B_w}\|\cdot\|_{\bar{p}}$. We also have $\|\cdot\|_{\bar{p}} \leq \|\cdot\|_1 \leq d^{1-1/\bar{p}}\|\cdot\|_{\bar{p}} \leq 2\|\cdot\|_{\bar{p}}$. Thus, under these norms we have diameter bound $B^2 = 1$ and Lipschitz constant $L^2 \leq 4B_w^2 L_w^2 + 4B_\theta^2 L_\theta^2$ [NJLS09]. Further the strong convexity assumption needed by Theorem 1 is satisfied with $\kappa = \max\left\{\frac{1}{\bar{p}-1}, \frac{1}{\bar{q}-1}\right\}$. This is because for any $p > 1$, $\frac{1}{2}\|\cdot\|_p^2$ is $(p-1)$-strongly convex w.r.t. $\|\cdot\|_p$ [Bec17].

**Private Algorithm Satisfying Relative Accuracy.**    To apply Theorem 1, we must construct an algorithm satisfying relative accuracy and $(\epsilon, \delta)$-DP. For this, we use the stochastic mirror prox algorithm of [JNT11], Algorithm 2. This algorithm will also have application in our analysis of SVIs later on.

---

**Algorithm 2** Stochastic Mirror Prox

**Require:** Learning rate $\eta$, Operator oracle $\mathcal{O}$, Initial point $z_0 \in \mathcal{Z}$, Regularization function $\psi$ minimized at $z_0$, Iterations $T$
1: **for** $t = 1 \ldots T$ **do**
2:     $\tilde{z}_t = \arg\min_{u \in \mathcal{Z}} \{\psi(u) + \langle \eta\mathcal{O}(z_{t-1}) - \nabla\psi(z_{t-1}), u\rangle\}$
3:     $z_t = \arg\min_{u \in \mathcal{Z}} \{\psi(u) + \langle \eta\mathcal{O}(\tilde{z}_t) - \nabla\psi(z_{t-1}), u\rangle\}$
4: **end for**
5: **SSP output:** $\frac{1}{T}\sum_{t=1}^T z_t$
6: **SVI output:** $z_{t^*}$ **for** $t^* \sim \mathsf{Unif}([T])$

---

This algorithm takes as input a stochastic oracle for the saddle operator of $f$, $\mathcal{O}$, and a strongly convex regularizer, $\psi$. We leverage this algorithm by constructing a differentially private version of the operator oracle $\mathcal{O}$, and taking as output the average iterate $\bar{z} = \frac{1}{T}\sum_{t=1}^T z_t$. We make the oracle private by adding Gaussian noise to minibatch estimates of the saddle operator. It is then easy to show the whole algorithm is private by composition results and the post processing properties of differential privacy. We defer these details to Appendix C.2, and here state the final relative accuracy bound.

---
**Algorithm 3** Recursive Regularization for SVIs: $\mathcal{R}_{\text{SVI}}$
---
**Require:** Dataset $S$, monotone operator $g$, Subroutine $\mathcal{A}_{\text{emp}}$, regularity parameter $\kappa > 0$, regulariza-
   tion parameter $\lambda \geq \frac{L\sqrt{\kappa}}{B\sqrt{n}}$, constraint set diameter $B$, Strongly monotone operator $\rho$
 1: Let $n' = n/\log_2(n)$, and $T = \log_2(\frac{L}{\kappa B \lambda})$.
 2: Let $S_1, ..., S_T$ be a disjoint partition of $S$ with each $S_t$ of size $n'$ *(always possible due to the*
   *condition on $\lambda$)*
 3: $\bar{z}_0$ be any point in $\mathcal{Z}$
 4: Define function $(z, x) \mapsto g^{(1)}(z; x) = g(z; x) + 2\lambda \cdot \rho(z - \bar{z}_0)$
 5: **for** $t = 1$ to $T$ **do**
 6:    $\bar{z}_t = \mathcal{A}_{\text{emp}}\left(S_t, g^{(t)}, \bar{z}_{t-1}, \frac{B}{2^t}\right)$
 7:    Define function $(z, x) \mapsto g^{(t+1)}(z; x) = g^{(t)}(z; x) + 2^{t+1}\lambda \cdot \rho(z - \bar{z}_t)$
 8: **end for**
 9: **Output:** $\bar{z}_T$
---

**Lemma 3.** *Under the setup described above, there exists an $(\epsilon, \delta)$-DP algorithm which sat-*
*isfies $\hat{\alpha}$-relative accuracy with parameter $\hat{\alpha} = O\left(\sqrt{\kappa(B_w^2 L_w^2 + B_\theta^2 L_\theta^2)}\left(\frac{\sqrt{d\log(1/\delta)\tilde{\kappa}}}{n\epsilon} + \frac{1}{\sqrt{n}}\right)\right)$*
*and runs in $O\left(\min\left\{\frac{\sqrt{\kappa}n^2\epsilon^{1.5}}{\log^2(n)\sqrt{d\log(1/\delta)\tilde{\kappa}}}, \frac{\sqrt{\kappa}n^{3/2}}{\log^{3/2}(n)}\right\}\right)$ number of gradient evaluations, where $\tilde{\kappa} =$*
$1 + \mathbf{1}\{p < 2 \vee q < 2\} \cdot \log(d)$.

**Main Result for DP $\ell_p/\ell_q$ SSPs.**    Applying now the result of recursive regularization, Theorem
1, we obtain the optimal rate for $\ell_p/\ell_q$ SSPs (up to logarithmic factors). Recall under our chosen
norm that $B \leq 1$ and $L^2 \leq 4B_w^2 L_w^2 + 4B_\theta^2 L_\theta^2$. Further, the privacy of Algorithm 1 follows from the
privacy of $\mathcal{A}_{\text{emp}}$ and the parallel composition and post processing properties of differential privacy.

**Corollary 1.** *There exists an Algorithm, $\mathcal{R}$, which is $(\epsilon, \delta)$-DP, has number of gradient evaluations*
*bounded by $O\left(\min\left\{\frac{\sqrt{\kappa}n^2\epsilon^{1.5}}{\log(n)\sqrt{d\log(1/\delta)\tilde{\kappa}}}, \frac{\sqrt{\kappa}n^{3/2}}{\sqrt{\log(n)}}\right\}\right)$, and satisfies (up to $\log(n)$ factors),*

$$\text{Gap}_{\text{SP}}(\mathcal{R}) = \tilde{O}\left(\kappa^{2.5}\sqrt{B_w^2 L_w^2 + B_\theta^2 L_\theta^2}\left(\frac{\sqrt{d\log(1/\delta)\tilde{\kappa}}}{n\epsilon} + \frac{1}{\sqrt{n}}\right)\right).$$

*($\kappa$ is at most $\log(d)$.)*

Note that in the $\ell_2/\ell_2$-setting $\kappa = 1$ and the above exactly recovers the result of [BGM23]. We further
recall that the near optimality of this result is established by existing lower bounds for stochastic
minimization, which is a special case of SSPs [BFTT19, BGN21].

## 5   Extension to Variational Inequalities

In this section, we start by discussing the modifications that must be made to the recursive regulariza-
tion algorithm to handle the more general structure of SVIs. We then discuss key ideas in the analysis
and how to apply the algorithm to SVIs in the $\ell_p$ setting. We recall that we here assume the operator
$g$ is monotone, $L$-bounded and $\beta$-Lipschitz.

### 5.1   Recursive Regularization Algorithm for SVIs

Algorithm 3 bears many similarities to Algorithm 1. Most notable among the differences is that we
here regularize with a strongly monotone operator $\rho$ instead of a strongly-convex/strongly-concave
function. In our eventual application we will use $\rho = \nabla(\frac{1}{2}\|\cdot\|^2)$, but strictly we only require that $\rho$
satisfies the following.

**Assumption 1.** *For $\kappa > 0$, $\rho: \mathcal{Z} \mapsto \mathbb{R}^d$ is $\frac{1}{\kappa}$-strongly monotone w.r.t. $\|\cdot\|$ and satisfy $\|\rho(z)\|_* \leq \|z\|$*
*for all $z \in \mathcal{Z}$.*

When $\rho = \nabla(\frac{1}{2}\|\cdot\|^2)$, the second half of the condition is always guaranteed by the properties of the
dual norm (see Fact 2). When $\rho$ satisfies Assumption 1, we obtain the following guarantee.

**Theorem 2.** *Let $\mathcal{A}_{\text{emp}}$ satisfy $\hat{\alpha}$-relative stationarity for any $(5L)$-bounded monotone operator and dataset of size $n' = \frac{n}{\log(n)}$. Let $\rho$ satisfy Assumption 1. Then Algorithm 1, run with $\mathcal{A}_{\text{emp}}$ as a subroutine and $\lambda = \frac{48}{B}\left(\hat{\alpha}\kappa^3 + \frac{(\beta B + L)\kappa^2}{\sqrt{n'}}\right)$, satisfies*

$$\text{Gap}_{\text{VI}}(\mathcal{R}_{SVI}) = O\left(\log(n)B\hat{\alpha}\kappa^3 + \frac{\log^{3/2}(n)B(\beta B + L)\kappa^2}{\sqrt{n}}\right).$$

The full proof is in Appendix B.2. We discuss the key ideas in the following subsection.

## 5.2 Analysis Idea

Unfortunately, the analysis used for stochastic saddle point problems does not easily extend to variational inequalities due to the fact that the VI-gap and SP-gap behave in fundamentally different ways. Even though SSPs are a special case of SVIs (when the operator in question is the saddle operator of the loss function), a bound on the VI-gap does not imply a bound on the SSP-gap. Further, a natural attempt to extend the notion of relative accuracy (Definition 3) to SVIs by asking $\mathcal{A}_{\text{emp}}$ to bound $\langle G_{\mathcal{D}}(z_S^*), \mathcal{A}(S) - z_S^* \rangle$ does not work, because such a term does not yield an upper bound on the distance $\|\mathcal{A}(S) - z_S^*\|$. A similar problem holds for generalization measures in Eqn. (6).

**A New Empirical Accuracy Measure.** Motivated by the above issues, we introduce a new relative accuracy measure for our analysis of SVIs. Importantly, this notion will allow us to bound the distance between the output of $\mathcal{A}_{\text{emp}}$ and the empirical equilibrium point of the strongly monotone operator created at each round of the recursive regularzation algorithm.

**Definition 3** ($\hat{\alpha}$-relative stationarity)**.** *Given a dataset $S' \in \mathcal{X}^{n'}$, operator $g'$, and an initial point $z'$, we say that $\mathcal{A}_{\text{emp}}$ satisfies $\hat{\alpha}$-relative stationarity w.r.t. to the empirical equilibrium $z_{S'}^*$ of $G_{S'}(z) = \frac{1}{n'}\sum_{x \in S'} g'(z; x)$, if, $\forall \hat{D} > 0$, whenever $\mathbb{E}\left[\|z' - z_{S'}^*\|\right] \leq \hat{D}$, the output $\bar{z}$ of $\mathcal{A}_{\text{emp}}$ satisfies $\mathbb{E}\left[\langle G(\bar{z}), \bar{z} - z_{S'}^* \rangle\right] \leq \hat{D}\hat{\alpha}$.*

**Modified Generalization Measure.** The generalization measure we use can be modified in a similar fashion.

**Lemma 4.** *Let $g(z; x) = g_1(z; x) + g_2(z)$ such that $g_1 : \mathcal{Z} \times \mathcal{X} \mapsto \mathbb{R}^d$ is $L$-bounded and $\beta$-Lipschitz with respect to $z$ and $g_2 : \mathcal{Z} \mapsto \mathbb{R}^d$ is any (data indepedent) operator. Let $z^* \in \mathcal{Z}$ be its population equilibrium point. For any $x \in \mathcal{X}$, define $h(z; x) = \langle g(z; x), z - z^* \rangle$. For $S \sim \mathcal{D}^n$, let $H_S(z) = \frac{1}{n}\sum_{x \in S} h(z; x)$ and $H_{\mathcal{D}}(z) = \mathbb{E}_{x \sim \mathcal{D}}[h(z; x)]$. Then for any $\Delta$-UAS algorithm, $\mathcal{A}$, one has $\mathbb{E}_{S, \mathcal{A}}\left[H_{\mathcal{D}}(\mathcal{A}(S)) - H_S(\mathcal{A}(S))\right] \leq \Delta(\beta B + L)$.*

The proof is similar to that of Lemma 2, but must account for additional complications. First, the function $h$ may not be Lipschitz if the regularizer, represented by $g_2$ above, is not Lipschitz. This happens, for example, in the $\ell_1$ setting. Thus, we must decompose $h$ in the stability-implies-generalization analysis and handle the non-Lipschitz, but data-independent, term $g_2$ separately. Then, the Lipschitzness of the remainder is established using that fact that $g_1$ is both bounded and Lipschitz. The full proof is in Appendix D.1.

## 5.3 Application to DP variational inequalities in the $\ell_p$ setting

In order to apply Theorem 2 to SVIs with an $\ell_p$ setup, we pick $\|\cdot\| = \frac{1}{B}\|z\|_{\bar{p}}$ where $\bar{p} = \max\{p, 1 + \frac{1}{\log(d)}\}$ [2]. We will use the regularizer,

$$\rho(z) = \nabla(\frac{1}{2B}\|z\|_{\bar{p}}^2). \tag{7}$$

Note the above operator is uniquely defined since $\bar{p} > 1$, and thus $\|\cdot\|_{\bar{p}}^2$ is differentiable [Gui09]. This choice of $\rho$ satisfies Assumption 1 with parameter $\kappa = \frac{1}{\bar{p}-1}$. To see this, first observe that

---
[2]In contrast to our results on SSPs, rescaling by $\frac{1}{B}$ is not necessary in this case, but we do so to maintain consistency.

for any $\bar{p} > 1$, $\frac{1}{2}\|\cdot\|_{\bar{p}}^2$ is $(\bar{p}-1)$-strongly convex w.r.t. $\|\cdot\|_{\bar{p}}$ and the gradient operator of a differentiable $\mu$-strongly convex function is $\mu$-strongly monotone. Second, for any norm and it's dual $\|\nabla(\frac{1}{2}\|z\|^2)\|_* \leq \|z\|$ for all $z$ (see Fact 2 in Appendix A).

To obtain relative stationarity guarantees, we again apply Algorithm 2 with a differentially private oracle for the empirical operator $G_S$. The proof falls out of our existing analysis for stochastic mirror prox given in Appendix C.2. Specifically, Theorem 4 in the case where $\Theta$ is the empty set implies there exists an $(\epsilon, \delta)$-DP implementation of mirror prox which satisfies $\hat{\alpha}$-relative stationarity with

$$\hat{\alpha} = O\Big(\frac{\sqrt{\kappa}BL\sqrt{d\log(1/\delta)(1 + \mathbf{1}\{p < 2\} \cdot \log(d))}}{n\epsilon} + \frac{\sqrt{\kappa}BL}{\sqrt{n}}\Big).$$

We here highlight one notable difference that arises with the algorithm. Typically, after running an algorithm like stochastic mirror prox for $t$ iterations, one obtains a bound on the quantity $\mathbb{E}\left[\frac{1}{T}\sum_{t=1}^{T}\langle G_S(z_t), z_t - z\rangle\right]$. Analysis then proceeds to bound the (empirical) VI-gap of the average iterate by leveraging monotonicity of the operator. However, this application of monotonicity does not help for the purposes of relative stationarity. For this reason, we instead select an iterate uniformly at random. We note this step is nonstandard as such a selection does not necessarily yield any bound on the (empirical) VI-gap.

**Main Result for $\ell_p$ DP SVIs.** Using Algorithm 3 and the implementation of $\mathcal{A}_{\mathsf{emp}}$ described above, we ultimately obtain the following result as a corollary of Theorem 2 and Theorem 4. Recall under our choice of norm we have diameter bound 1 and operator bound $BL$. Further, we can leverage existing lower bounds to show that this rate is near optimal; more details are available in Appendix E.

**Corollary 2.** *There exists an Algorithm, $\mathcal{R}$, which is $(\epsilon, \delta)$-DP, has gradient evaluations bounded by* $O\big(\min\big\{\frac{\sqrt{\kappa}n^2\epsilon^{1.5}}{\log(n)\sqrt{d\log(1/\delta)\tilde{\kappa}}}, \frac{\sqrt{\kappa}n^{3/2}}{\sqrt{\log(n)}}\big\}\big)$*, and satisfies (up to $\log(n)$ factors)*

$$\mathrm{Gap}_{\mathsf{VI}}(\mathcal{R}) = \tilde{O}\Big(\kappa^{3.5}BL\big(\frac{\sqrt{d\log(1/\delta)\tilde{\kappa}}}{n\epsilon} + \frac{1}{\sqrt{n}}\big)\Big). \tag{8}$$

*where $\kappa = \frac{1}{\max\{p, 1+\frac{1}{\log(d)}\}-1}$ is at most $\log(d)$ and $\tilde{\kappa} = 1 + \mathbf{1}\{p < 2\} \cdot \log(d)$.*

**Near Linear Time Algorithm for the $\ell_2$ Setting.** Because we assume the operator is Lipschitz, in the $\ell_2$ setting, we can leverage existing accelerated optimization techniques to achieve a near linear time version of $\mathcal{A}_{\mathsf{emp}}$, in a similar fashion to [ZTOH22, BGM23]. Specifically, using the accelerated SVRG algorithm of [PB16] and Gaussian noise it is possible to obtain the rate in Eqn. (8) (with $\kappa = \tilde{\kappa} = 1$) in $O(n + \beta n \log(n/\delta))$ gradient evaluations. We provide full details in Appendix D.3.

## Acknowledgments and Disclosure of Funding

R. Bassily's and M. Menart's research was supported by NSF CAREER Award 2144532 and, in part, by NSF Award 2112471. C. Guzmán's research was partially supported by INRIA Associate Teams project, ANID FONDECYT 1210362 grant, ANID Anillo ACT210005 grant, and National Center for Artificial Intelligence CENIA FB210017, Basal ANID.

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

## A Supplementary Lemmas

**Fact 2.** *For any $z$, it holds that $\|\nabla(\frac{1}{2}\|z\|^2)\|_* \leq \|z\|$.*

*Proof.* By the chain rule we have $\nabla(\frac{1}{2}\|z\|^2) = \|z\| \cdot \nabla(\|z\|)$ and so $\|\nabla(\frac{1}{2}\|z\|^2)\|_* = \|z\| \cdot \|\nabla(\|z\|)\|_*$. Thus, it only remains to show that $\|\nabla(\|z\|)\|_* \leq 1$. By the definition of the dual norm we have $\|\nabla(\|z\|)\|_* = \max_{v:\|v\|\leq 1} \{\langle \nabla(\|z\|), v \rangle\}$. Further, by the definition of the subgradient we have for any $z'$ that $\|z'\| - \|z\| \geq \langle \nabla(\|z\|), z' - z \rangle$. Substituting $v = z' - z$, we have

$$\langle \nabla(\|z\|), v \rangle \leq \|v + z\| - \|z\| \leq \|v\| + \|z\| - \|z\| = \|v\| \leq 1,$$

as desired. $\square$

**Lemma 5.** *Let $\mu, \lambda > 0$ and $\rho$ be $\mu$-strongly monotone w.r.t. $\|\cdot\|$. Then for any point $z_0 \in \mathcal{Z}$, the algorithm which outputs the unique equilibrium of $G_S(z) + \frac{\lambda}{\mu}(\rho(z) - \rho(z_0))$ is $\left(\frac{2L}{\mu\lambda n}\right)$-uniform argument stable w.r.t. $S$.*

Note without loss of generality we can always choose $z_0$ to be the point such that $\rho(z_0) = \mathbf{0}$, which is guaranteed to exist since $\rho$ is strongly monotone. Thus this result also holds for regularization of the form $G_S(z) + \lambda \cdot \rho(z)$. Further, since the saddle operator of a strongly-convex/strongly-concave loss is strongly monotone, and the resulting SSP shares its equilibrium point with the corresponding SVI, Lemma 1 given in the preliminaries is also established through this result.

*Proof.* Let $z_\lambda$ and $z'_\lambda$ denote the equilibrium points of the regularized operators w.r.t. to adjacent datasets $S$ and $S'$ respectively. By the equilibrium condition we have for any $z \in \mathcal{Z}$

$$\langle G_S(z_\lambda) + \lambda[\rho(z_\lambda) - \rho(z_0)], z - z_\lambda \rangle \geq 0$$
$$\langle G_{S'}(z'_\lambda) + \lambda[\rho(z'_\lambda) - \rho(z_0)], z - z'_\lambda \rangle \geq 0$$
$$\implies \langle G_S(z_\lambda) - G_{S'}(z'_\lambda) + \lambda[\rho(z_\lambda) - \rho(z'_\lambda)], z'_\lambda - z_\lambda \rangle \geq 0.$$

Since $\rho$ is a 1-strongly monotone operator we have

$$\langle G_S(z_\lambda) - G_{S'}(z'_\lambda), z'_\lambda - z_\lambda \rangle \geq \lambda \langle \rho(z_\lambda) - \rho(z'_\lambda), z_\lambda - z'_\lambda \rangle \geq \mu\lambda \|z_\lambda - z'_\lambda\|^2.$$

Now we can use the monotonicity of $G$ to derive,

$$\begin{aligned}
\mu\lambda\|z_\lambda - z'_\lambda\|^2 &\leq \langle G_S(z_\lambda) - G_{S'}(z'_\lambda), z'_\lambda - z_\lambda \rangle \\
&= \langle G_S(z_\lambda) - G_{S'}(z_\lambda), z'_\lambda - z_\lambda \rangle + \langle G_{S'}(z_\lambda) - G_{S'}(z'_\lambda), z'_\lambda - z_\lambda \rangle \\
&\overset{(i)}{\leq} \langle G_S(z_\lambda) - G_{S'}(z_\lambda), z'_\lambda - z_\lambda \rangle \\
&\overset{(ii)}{\leq} \|G_S(z_\lambda) - G_{S'}(z_\lambda)\|_* \cdot \|z'_\lambda - z_\lambda\| \\
&\overset{(iii)}{\leq} \frac{2L}{n}\|z'_\lambda - z_\lambda\|.
\end{aligned}$$

Above $(i)$ comes from the monotonicity of $G$, step $(ii)$ comes from Hölder's inequality, and step $(iii)$ comes from the fact that $S$ and $S'$ differ in at most one point. Simple algebra now obtains $\|z_\lambda - z'_\lambda\| \leq \frac{2L}{\mu\lambda n}$. $\qquad\square$

**Fact 3.** *There exists a differentiable convex-concave loss, distribution $\mathcal{D}$, and point $z'$ such that when the operator is the saddle operator of the loss, $\mathrm{Gap}_{\mathsf{VI}}(z) < \mathrm{Gap}_{\mathsf{SP}}(z)$.*

*Proof.* We show that the VI-gap is upper bounded by the excess risk of some convex function with operator $g(z) = \nabla f(z)$. Since stochastic convex optimization is a special case of SSPs, this shows there exist scenarios where $\mathrm{Gap}_{\mathsf{SP}} \not\leq \mathrm{Gap}_{\mathsf{VI}}$. Specifically, when the dual player space $\Theta$ is singleton, the SP-gap becomes the excess risk.

Let $f(z; x) = z^2$ be defined over $z \in [0, 1]$, and so it does not matter what the distribution or data domain is. Note that $\mathrm{Gap}_{\mathsf{VI}}$ becomes

$$\mathrm{Gap}_{\mathsf{VI}}(z) = \max_{u \in [0,1]} \left\{ \langle \nabla f(u), z - u \rangle \right\} = \max_{u \in [0,1]} \left\{ 2uz - 2u^2 \right\}.$$

Note that in this case the SP-gap is just the excess risk, $\mathrm{Gap}_{\mathsf{SP}}(z) = z^2 - \min_{z \in [0,1]} \left\{ z^2 \right\} = z^2$.

Consider the point $z = 1$. It is easy to show that $\mathrm{Gap}_{\mathsf{VI}}(1) = 0.5$ but $\mathrm{Gap}_{\mathsf{SP}}(1) = 1$, proving the claim. $\qquad\square$

# B  Missing Results from Section 3

## B.1  Proof of Lemma 2

**Lemma 6.** *(Restatement of Lemma 2) Let $f : \mathcal{Z} \times \mathcal{X} \mapsto \mathbb{R}$ be $L$-Lipschitz. Let $[w^*, \theta^*] \in \mathcal{Z}$ be the population saddle point. For any $x \in \mathcal{X}$ define $h([w, \theta]; x) = f(w, \theta^*; x) - f(w^*, \theta; x)$. For $S \sim \mathcal{D}^n$, let $H_S(z) = \frac{1}{n} \sum_{x \in S} h(z; x)$ and $H_{\mathcal{D}}(z) = \underset{x \sim \mathcal{D}}{\mathbb{E}}[h(z; x)]$. Then for any $\Delta$-UAS algorithm, $\mathcal{A}$, one has $\underset{S, A}{\mathbb{E}}[H_{\mathcal{D}}(\mathcal{A}(S)) - H_S(\mathcal{A}(S))] \leq 2\Delta L$.*

*Proof.* This result follows simply from two facts. First, because $f$ is an $L$-Lipschitz function for any $x \in \mathcal{X}$, we can show that $h$ is $L$-Lipschitz. For any $[w, \theta], [w', \theta'] \in \mathcal{Z}$ observe,

$$\begin{aligned}
h([w, \theta]) - h([w', \theta']) &= [f(w, \theta^*; x) - f(w^*, \theta; x)] - [f(w', \theta^*; x) - f(w^*, \theta'; x)] \\
&= f(w, \theta^*; x) - f(w', \theta^*; x) + f(w^*, \theta'; x) - f(w^*, \theta; x) \\
&\leq 2L\|[w, \theta] - [w', \theta']\|.
\end{aligned}$$

The rest of the proof essentially follows from the stability implies generalization proof of [BE02] since $H_{\mathcal{D}}$ and $H_S$ have a statistical form w.r.t. $h$. In more detail, for any $i \in [n]$ denote $S^{(i)}$ as the dataset which replaces the $i$'th datapoint of $S$, $x_i$, with a fresh sample from $\mathcal{D}$, $x'$. We have the following:

$$
\begin{aligned}
\mathbb{E}_{S,\mathcal{A}} \left[ H_{\mathcal{D}}(\mathcal{A}(S)) - H_S(\mathcal{A}(S)) \right] &= \mathbb{E}_{S,\mathcal{A}} \left[ \mathbb{E}_x \left[ h(\mathcal{A}(S); x) \right] - \frac{1}{n} \sum_{x \in S} [h(\mathcal{A}(S); x)] \right] \\
&= \mathbb{E}_{S, x' \sim \mathcal{D}^{n+1}, i \sim \mathsf{Unif}([n])} \left[ h(\mathcal{A}(S^{(i)}); x_i) - h(\mathcal{A}(S); x_i) \right] \\
&= \mathbb{E} \left[ h(\mathcal{A}(S^{(i)}; x_i) - h(\mathcal{A}(S); x_i) \right] \\
&\leq \mathbb{E} \left[ L \| \mathcal{A}(S^{(i)}) - \mathcal{A}(S) \| \right] \leq 2L\Delta.
\end{aligned}
$$

The last step follows from the previously established Lipschitzness property of $h$. $\qquad \square$

## B.2 Convergence of Recursive Regularization for SSPs

We will first prove the following more general version statement of Theorem 1.

**Theorem 3.** *Let* $\lambda \geq \frac{48 L \kappa^{3/2}}{B\sqrt{n'}}$ *and* $\mathcal{A}_{\mathsf{emp}}$ *be such that for all* $t \in [T]$ *it holds that* $\mathbb{E}\left[ \left\| \bar{z}_t - z_{S,t}^* \right\|^2 \right] \leq \frac{B^2}{12 \cdot 2^{2t} \kappa^{3/2}}$. *Then Recursive Regularization satisfies*

$$
\mathrm{Gap}_{\mathsf{SP}}(\mathcal{R}_{SSP}) = O\left( \log(n) B^2 \lambda \right).
$$

To prove this result, it will be helpful to first show several intermediate results. We start by defining several useful quantities. Define $\{\mathcal{F}_t\}_{t=0}^T$ as the filtration where $\mathcal{F}_t$ is the sigma algebra induced by all randomness up to $\bar{z}_t$. For notational convenience we denote $f^{(0)}(w, \theta; x) = f(w, \theta; x)$. Then for every $t \in \{0, 1, ..., T\}$ we define

- $z_t^* = [w_t^*, \theta_t^*]$ : saddle point of $F_{\mathcal{D}}^{(t)}(w, \theta) := \mathbb{E}_{x \sim \mathcal{D}} \left[ f^{(t)}(w, \theta; x) \right]$;

- $z_{S,t}^* = [w_{S,t}^*, \theta_{S,t}^*]$ : saddle point of $F_S^{(t)}(w, \theta) := \frac{1}{n'} \sum_{x \in S_t} f^{(t)}(w, \theta; x)$;

- $H_{\mathcal{D}}^{(t)}([w, \theta]) := F_{\mathcal{D}}^{(t)}(w, \theta_t^*) - F_{\mathcal{D}}^{(t)}(w_t^*, \theta)$ : the relative accuracy function w.r.t. the population loss and population saddle point; and,

- $H_S^{(t)}([w, \theta]) := F_S^{(t)}(w, \theta_t^*) - F_S^{(t)}(w_t^*, \theta)$ : the relative accuracy function w.r.t. the empirical loss and population saddle point

We recall the generalization properties of $H_S^{(t)}$ and $H_{\mathcal{D}}^{(t)}$ shown in Lemma 6. Crucially, the power of $H_{\mathcal{D}}^{(t)}$ is that it bounds the distance of a point to $z_t^*$.

**Fact 4** ([ZHWZ21], Theorem 1). *Let* $F : \mathcal{Z} \mapsto \mathbb{R}^d$ *be a* $\gamma$-SC/SC *function and let* $[w^*, \theta^*]$ *be the saddle point. Then* $\| [w, \theta] - [w^*, \theta^*] \|^2 \leq \frac{2(F(w, \theta^*) - F(\theta^*, w))}{\lambda}$.

We now establish two distance inequalities which will be used when analyzing the final gap bound in Theorem 3. The first inequality below bounds the distance of the output of the $t$-th round to the equilibrium of $G_{\mathcal{D}}^{(t)}$. The second inequality bounds how far the population equilibrium moves after another regularization term is added.

**Lemma 7.** *Assume the conditions of Theorem 3 hold. Then for every* $t \in [T]$, *the following holds*

**P.1** $\mathbb{E}\left[ \| \bar{z}_t - z_t^* \| \right]^2 \leq \mathbb{E}\left[ \| \bar{z}_t - z_t^* \|^2 \right] \leq \frac{B^2}{2^{2t} \kappa}$; *and,*

**P.2** $B_t^2 := \mathbb{E}\left[ \| z_t^* - z_{t-1}^* \| \right]^2 \leq \mathbb{E}\left[ \| z_t^* - z_{t-1}^* \|^2 \right] \leq \frac{B^2}{2^{2(t-1)}}$.

We note that in contrast to [BGM23] and other analyses of recursive regularization, we define Property **P.2** to bound $\mathbb{E}\left[\left\|z_t^* - z_{t-1}^*\right\|\right]$ instead of $\mathbb{E}\left[\left\|z_t^* - \bar{z}_{t-1}\right\|\right]$. In [BGM23], the latter choice led to a need to bound $\mathbb{E}\left[\widehat{\mathrm{Gap}}_{\mathsf{SP}}(\bar{z}_{t-1})\right]$ for all $t \in [T]$, which our analysis avoids. In particular, one can observe how the derivation of Eqn. (11) in the proof changes when $z_{t-1}^*$ is replaced with $\bar{z}_{t-1}$.

*Proof of Lemma 7.* We will prove both properties via induction on $B_1, ..., B_T$. Specifically, for each $t \in [T]$ we will introduce two terms $E_t$ and $F_t$,, and show that these terms are bounded if the bound on $B_t$ holds and that $B_t$ holds if $E_{t-1}$ and $F_{t-1}$ are bounded. Property **P.1** is then established as a result of the fact that $\mathbb{E}\left[\left\|\bar{z}_t - z_t^*\right\|^2\right] \leq 2(E_t + F_t)$. Note that $B_1$ holds as the base case because $\mathbb{E}\left[\left\|z_1^* - z_0^*\right\|^2\right] \leq B^2$.

**Property P.1:** We here prove that if $B_t$ is sufficiently bounded, then $E_t$ and $F_t$ are bounded where for $t \in [T]$ we define

$$E_t = \mathbb{E}\left[\left\|\bar{z}_t - z_{S,t}^*\right\|^2\right], \qquad\qquad F_t = \frac{\kappa}{2^t \lambda} \mathbb{E}\left[H_{\mathcal{D}}^{(t)}\left(z_{S,t}^*\right)\right]. \qquad (9)$$

Additionally, this will establish property **P.1** because for any $t \in [T]$ it holds that,

$$\mathbb{E}\left[\left\|\bar{z}_t - z_t^*\right\|^2\right] \leq 2\left(\mathbb{E}\left[\left\|\bar{z}_t - z_{S,t}^*\right\|^2\right] + \mathbb{E}\left[\left\|z_{S,t}^* - z_t^*\right\|^2\right]\right)$$

$$\leq 2\left(\underbrace{\mathbb{E}\left[\left\|\bar{z}_t - z_{S,t}^*\right\|^2\right]}_{E_t} + \underbrace{\frac{\kappa}{2^t \lambda}\mathbb{E}\left[H_{\mathcal{D}}^{(t)}\left(z_{S,t}^*\right)\right]}_{F_t}\right). \qquad (10)$$

The second inequality comes from the strong monotonicity of the operator (see Fact 4).

Note that $E_t$ is bounded by the assumption made in the statement of the theorem statement. We thus turn our attention towards bounding $F_t$. We have

$$\frac{\kappa}{2^t\lambda}\mathbb{E}\left[H_{\mathcal{D}}^{(t)}\left(z_{S,t}^*\right)\right] = \frac{\kappa}{2^t\lambda}\,\mathbb{E}\left[\mathbb{E}\left[H_{\mathcal{D}}^{(t)}\left(z_{S,t}^*\right)\Big|\mathcal{F}_{t-1}\right]\right]$$

$$\overset{(i)}{\leq} \frac{\kappa}{2^t\lambda}\left(\mathbb{E}\left[\mathbb{E}\left[H_S^{(t)}\left(z_{S,t}^*\right)\Big|\mathcal{F}_{t-1}\right]\right] + \frac{\kappa L^2}{2^t\lambda n'}\right)$$

$$= \frac{\kappa}{2^t\lambda}\left(\mathbb{E}\left[\mathbb{E}\left[F_S^{(t)}(w_{S,t}^*,\theta_t^*) - F_S^{(t)}(w_t^*,\theta_{S,t}^*)\Big|\mathcal{F}_{t-1}\right]\right] + \frac{2\kappa L^2}{2^t\lambda n'}\right)$$

$$\overset{(ii)}{=} \frac{2\kappa^2 L^2}{2^{2t}\lambda^2 n'} \leq \frac{B^2}{1152 \cdot 2^{2t}\kappa}.$$

Inqequality $(i)$ comes from the fact that stability implies generalization for $H^{(t)}$, Lemma 2. Note the algorithm which outputs this exact equilibrium point is $\frac{L^2}{2^t\lambda n'}$ stable (see Lemma 5/Assumption 1). Step $(ii)$ uses the fact that $z_{S,t}^*$ is the exact saddle point of the regularized objective, and so for any $[w,\theta] \in \mathcal{Z}$, $F_S^{(t)}(w_{S,t}^*,\theta) - F_S^{(t)}(w,\theta_{S,t}^*) \leq 0$. The final inequality uses the setting of $\lambda$.

We thus have a final bound $2(E_t + F_t) \leq \frac{B^2}{2^{2t}}$.

**Property P.2:** Now assume $B_{t-1}$ holds. We have

$$\mathbb{E}\left[\left\|z_t^* - z_{t-1}^*\right\|^2\right] \le \mathbb{E}\left[\frac{\kappa}{2^t\lambda}H_{\mathcal{D}}^{(t)}(z_{t-1}^*)\right]$$

$$= \mathbb{E}\left[\frac{\kappa}{2^t\lambda}\left(F_{\mathcal{D}}^{(t)}(w_{t-1}^*, \theta_t^*) - F_{\mathcal{D}}^{(t-1)}(w_t^*, \theta_{t-1}^*)\right)\right]$$

$$= \mathbb{E}\Big[\frac{\kappa}{2^t\lambda}\left(F_{\mathcal{D}}^{(t-1)}(w_{t-1}^*, \theta_t^*) - F_{\mathcal{D}}^{(t-1)}(w_t^*, \theta_{t-1}^*)\right)$$

$$+ \kappa\left(\|w_{t-1}^* - \bar{w}_{t-1}\|_w^2 - \|\theta_t^* - \bar{\theta}_{t-1}\|_\theta^2 - \|w_t^* - \bar{w}_{t-1}\|_w^2 + \|\theta_t^* - \bar{\theta}_{t-1}\|_\theta^2\right)\Big]$$

$$\overset{(i)}{\le} \mathbb{E}\left[\kappa\|z_{t-1}^* - \bar{z}_{t-1}\|^2\right] \tag{11}$$

Inequality $(i)$ above comes from removing negative terms and the fact that $z_{t-1}^*$ is the saddle point w.r.t. $F_{\mathcal{D}}^{(t-1)}$. Using the induction argument we then complete the bound with the following:

$$\mathbb{E}\left[\left\|z_t^* - z_{t-1}^*\right\|^2\right] \le \mathbb{E}\left[\kappa\|z_{t-1}^* - \bar{z}_{t-1}\|^2\right] \le \kappa(E_{t-1} + F_{t-1}) \le \frac{B^2}{2^{2t}}$$

$\square$

We now turn to analyzing the utility of the algorithm to complete the proof.

*proof of Theorem 3.* Using the fact that $\widehat{\mathrm{Gap}}_{\mathsf{SP}}$ is $\sqrt{2}L$-Lipschitz and property **P.1**, we have

$$\mathbb{E}\left[\widehat{\mathrm{Gap}}_{\mathsf{SP}}(\bar{z}_T) - \widehat{\mathrm{Gap}}_{\mathsf{SP}}(z_T^*)\right] \le \sqrt{2}L\mathbb{E}\left[\|\bar{z}_T - z_T^*\|\right]$$

$$\le \frac{\sqrt{2}BL}{2^T} \le \sqrt{2}B^2\lambda. \tag{12}$$

What remains is showing $\mathbb{E}\left[\widehat{\mathrm{Gap}}_{\mathsf{SP}}(w_T^*, \theta_T^*)\right]$ is $\tilde{O}(B\hat{\alpha} + \frac{BL}{\sqrt{n'}})$. Let $w' = \underset{\theta\in\Theta}{\arg\min}\, F_{\mathcal{D}}(w, \theta_T^*)$ and $\theta' = \underset{w\in\mathcal{W}}{\arg\max}\, F_{\mathcal{D}}(w_T^*, \theta)$. Using the fact that $F_{\mathcal{D}}$ is convex-concave we have

$$\widehat{\mathrm{Gap}}_{\mathsf{SP}}(w_T^*, \theta_T^*) = F_{\mathcal{D}}(w_T^*, \theta') - F_{\mathcal{D}}(w', \theta_T^*) \le \langle G_{\mathcal{D}}(w_T^*, \theta_T^*), [w_T^*, \theta_T^*] - [w', \theta']\rangle \tag{13}$$

where $G_{\mathcal{D}}$ is the population loss saddle operator. Further by the definition of $F^{(T)}$ and denoting $G_{\mathcal{D}}^{(T)}$ as the saddle operator for $F_{\mathcal{D}}^{(T)}$ we have

$$G_{\mathcal{D}}(w_T^*, \theta_T^*) = G_{\mathcal{D}}^{(T)}(w_T^*, \theta_T^*) - 2\lambda\sum_{t=0}^{T-1}2^{t+1}\nabla(\|[w_T^*, \theta_T^*] - [\bar{w}_t, \bar{\theta}_t]\|^2)$$

Thus plugging the above into Eqn. (13) we have

$$\widehat{\mathrm{Gap}}_{\mathsf{SP}}(w_T^*, \theta_T^*) \le \left\langle G_{\mathcal{D}}^{(T)}(w_T^*, \theta_T^*), [w_T^*, \theta_T^*] - [w', \theta']\right\rangle$$

$$- \left\langle 2\lambda\sum_{t=0}^{T-1}2^{t+1}\nabla(\|[w_T^*, \theta_T^*] - [\bar{w}_t, \bar{\theta}_t]\|^2), [w_T^*, \theta_T^*] - [w', \theta']\right\rangle$$

$$\le -\left\langle 2\lambda\sum_{t=0}^{T-1}2^{t+1}\nabla(\|[w_T^*, \theta_T^*] - [\bar{w}_t, \bar{\theta}_t]\|^2), [w_T^*, \theta_T^*] - [w', \theta']\right\rangle$$

$$\le 2B\lambda\sum_{t=0}^{T-1}2^{t+1}\left\|\nabla(\|[w_T^*, \theta_T^*] - [\bar{w}_t, \bar{\theta}_t]\|^2)\right\|_*$$

$$\le 2B\lambda\sum_{t=0}^{T-1}2^{t+1}\left\|[w_T^*, \theta_T^*] - [\bar{w}_t, \bar{\theta}_t]\right\|.$$

Above, the second inequality comes from the first order optimally conditions for $[w_T^*, \theta_T^*]$, the third from Cauchy Schwartz and a triangle inequality. The last inequality comes from the relationship between a norm and its dual, see Fact 2.

Taking the expectation on both sides of the above we have the following derivation,

$$\mathbb{E}\left[\widehat{\mathrm{Gap}}_{\mathsf{SP}}(w_T^*, \theta_T^*)\right] \le 2B\mathbb{E}\left[\lambda \sum_{t=0}^{T-1} 2^{t+1} \left\| [w_T^*, \theta_T^*] - [\bar{w}_t, \bar{\theta}_t] \right\| \right]$$

$$\overset{(i)}{\le} 4B\mathbb{E}\left[\lambda \sum_{t=0}^{T-1} 2^t \left( \left\| [w_t^*, \theta_t^*] - [\bar{w}_t, \bar{\theta}_t] \right\| + \sum_{r=t}^{T-1} \left\| [w_{r+1}^*, \theta_{r+1}^*] - [w_r^*, \theta_r^*] \right\| \right) \right]$$

$$= 4B\mathbb{E}\left[\lambda \sum_{t=0}^{T-1} 2^t \left\| [w_t^*, \theta_t^*] - [\bar{w}_t, \bar{\theta}_t] \right\| + \lambda \sum_{t=0}^{T-1} 2^t \sum_{r=t}^{T-1} \left\| [w_{r+1}^*, \theta_{r+1}^*] - [w_r^*, \theta_r^*] \right\| \right]$$

$$\overset{(ii)}{=} 4B\mathbb{E}\left[\lambda \sum_{t=0}^{T-1} 2^t \left\| [w_t^*, \theta_t^*] - [\bar{w}_t, \bar{\theta}_t] \right\| + \lambda \sum_{r=0}^{T-1} \sum_{t=0}^{r-1} 2^t \left\| [w_{r+1}^*, \theta_{r+1}^*] - [w_r^*, \theta_r^*] \right\| \right]$$

$$= 4B\mathbb{E}\left[\lambda \sum_{t=0}^{T-1} 2^t \left\| [w_t^*, \theta_t^*] - [\bar{w}_t, \bar{\theta}_t] \right\| + \lambda \sum_{r=0}^{T-1} \left\| [w_{r+1}^*, \theta_{r+1}^*] - [w_r^*, \theta_r^*] \right\| \sum_{t=0}^{r-1} 2^t \right]$$

$$\overset{(iii)}{\le} 4B\left(\lambda \sum_{t=0}^{T-1} 2^t \left(\frac{B}{2^t}\right) + \lambda \sum_{r=1}^{T-1} \left(\frac{B}{2^r}\right) \sum_{t=0}^{r-1} 2^t \right)$$

$$\le 4B\left(\lambda \sum_{t=0}^{T-1} 2^t \left(\frac{B}{2^t}\right) + \lambda \sum_{r=1}^{T-1} \left(\frac{B}{2^{r-1}}\right) \cdot (2^r - 1) \right)$$

$$= 4\lambda \sum_{t=0}^{T-1} B^2 + 8\lambda \sum_{r=1}^{T-1} B^2$$

$$\le 12T\lambda B^2 \tag{14}$$

Above, $(i)$ and the following inequality both come from the triangle inequality. Equality $(ii)$ is obtained by rearranging the sums. Inequality $(iii)$ comes from applying properties **P.1** and **P.2** proved above. The last equality comes from the setting of $\lambda$ and $T$.

Now using this result in conjunction with Eqn. (12) we have

$$\mathrm{Gap}_{\mathsf{SP}}(\mathcal{R}) = \sqrt{2}\lambda B^2 + 12T\lambda B^2 = O\left(\log(n)B^2\lambda\right).$$

Above we use the fact that $T = \log(\frac{L}{B\lambda})$ and $\lambda \ge \frac{L}{B\sqrt{n'}}$, and thus $T = O(\log(n))$. $\qquad\square$

Finally, we prove Theorem 1 leveraging the relative accuracy assumption.

*Proof of Theorem 1.* First, observe that under the setting of $\lambda = \frac{48}{B}\left(\hat{\alpha}\kappa^2 + \frac{L\kappa^{3/2}}{\sqrt{n'}}\right)$ used in the theorem statement that $\log(n)B^2\lambda = O\left(\log(n)B\hat{\alpha}\kappa^2 + \frac{\log^{3/2}(n)BL\kappa^{3/2}}{\sqrt{n}}\right)$. Thus what remains is to show that the distance condition required by Theorem 3 holds. That is, we now show that if $\mathcal{A}_{\mathsf{emp}}$ satisfies $\hat{\alpha}$-relative accuracy, then for all $t \in [T]$ it holds that $\mathbb{E}\left[\left\| \bar{z}_t - z_{S,t}^* \right\|^2\right] \le \frac{B^2}{12 \cdot 2^{2t}\kappa}$.

To prove this property, we must leverage the induction argument made by Lemma 7. Specifically, to prove the condition holds for some $t \in [T]$, assume $B_t^2 = \mathbb{E}\left[\left\| z_t^* - z_{t-1}^* \right\|\right]^2 \le \frac{B^2}{2^{2(t-1)}}$ (recall the base case for $t = 1$ trivially holds). As shown in the proof of Lemma 7, this implies that the quantities $E_t, F_t$ (as defined in 9) are bounded by $\frac{B^2}{1152 \cdot 2^{2t}}$. We thus have

$$\mathbb{E}\left[\left\| \bar{z}_t - z_{S,t}^* \right\|^2\right] \overset{(i)}{\le} \frac{\kappa\mathbb{E}\left[F_S^{(t)}(\bar{w}_t, \theta_{S,t}^*) - F_S^{(t)}(w_{S,t}^*, \bar{\theta}_t)\right]}{2^t\lambda} \overset{(ii)}{\le} \frac{2\kappa\hat{\alpha}B}{2^{2t}\lambda} \overset{(iii)}{\le} \frac{B^2}{12 \cdot 2^{2t}\kappa}, \tag{15}$$

where $B_t$ is as defined in property **P.2**. Inequality $(i)$ comes from the strong monotonicity of $G_S^{(t)}$, Fact 4. Inequality $(iii)$ comes from the setting $\lambda \ge 48\hat{\alpha}\kappa^2/B$. Inequality $(ii)$ comes from the $\hat{\alpha}$-relative accuracy assumption on $\mathcal{A}_{\mathsf{emp}}$, which holds so long as the expected distance is sufficiently bounded and each regularized loss is $(5L)$-Lipschitz. In this regard, note that

$$\mathbb{E}\left[\left\| z_{S,t}^* - \bar{z}_{t-1} \right\|\right] \le \mathbb{E}\left[\left\| z_{S,t}^* - z_t^* \right\| + \left\| z_t^* - z_{t-1}^* \right\| + \left\| z_{t-1}^* - \bar{z}_{t-1} \right\|\right]$$

$$\le (\sqrt{F_t} + B_t + \sqrt{E_{t-1}} + \sqrt{F_{t-1}}) \le \frac{B}{2^t}.$$

Further, each $f^{(t)}$ is $5L$-Lipschitz. We can see that,

$$\max_{z \in \mathcal{Z}} \|\nabla f^{(t)}(z, x)\|_* \le L + \|\sum_{k=0}^{t-1} 2^{k+1}\lambda\nabla(\|z - \bar{z}_t\|^2)\|_* \le L + \sum_{k=0}^{t-1} B2^{k+1}\lambda \le L + 4B2^T\lambda \le 5L.$$

$\square$

## C  Missing Results from Section 4

### C.1  General Guarantee for Stochastic Mirror Prox

We start with the follow general statement regarding the stochastic mirror prox algorithm applied to monotone operators. Notable for our purpose of solving SSPs is that the saddle operator of a convex-concave function is a monotone operator, but the following holds for any monotone operator.

**Lemma 8** (Implicit in [JNT11], Theorem 1)**.** *Let $\psi : \mathcal{Z} \mapsto \mathbb{R}$ be any non-negative function which is 1-strongly convex w.r.t. $\|\cdot\|$. Assume $\forall t \in [T]$ that $\mathbb{E}\left[\mathcal{O}(z_t)\right] = G(z_t)$ and $\mathbb{E}\left[\|\mathcal{O}(z_t) - G(z_t)\|_*^2\right] \le \tau^2$. Then for for any $z \in \mathcal{Z}$ Algorithm 2 satisfies*

$$\mathbb{E}\left[\frac{1}{T}\sum_{t=1}^{T}\langle G(z_t), z_t - z\rangle\right] = O\left(\frac{\mathbb{E}\left[\psi(z)\right]}{T\eta} + \frac{7\eta}{2}(L^2 + 2\tau^2)\right).$$

The above is slightly different than the statement in [JNT11], but can be easily extracted from their proof. Let $V_\psi$ denote the Bregman divergence w.r.t. $\psi$; i.e. $V_\psi(z, z') = \psi(z) - \psi(z') - \langle\nabla\psi(z'), z - z'\rangle$. Under our assumptions [JNT11, Eqn. (80)] gives

$$\mathbb{E}\left[\frac{1}{T}\sum_{t=1}^{T}\langle G(z_t), z_t - z\rangle\right] = O\left(\frac{\mathbb{E}\left[V_\psi(z, z_0)\right]}{T\eta} + \frac{7\eta}{2}(L^2 + 2\tau^2)\right).$$

Note that since $z_0$ is the minimizer of $\psi$, we have $V_\psi(z, z_0) \le \psi(z)$.

Before proving the relative accuracy guarantee of stochastic mirror prox, we also restate the following composition result for Gaussian mechanism known as the moments accountant.

**Lemma 9** ([ACG+16, KLL21])**.** *Let $\epsilon, \delta \in (0, 1]$ and $c$ be a universal constant. Let $D \in \mathcal{Y}^n$ be a dataset over some domain $\mathcal{Y}$, and let $h_1, ..., h_T : \mathcal{Y} \mapsto \mathbb{R}^d$ be a series of (possibly adaptive) queries such that for any $y \in \mathcal{Y}$, $t \in [T]$, $\|h_t(y)\|_2 \le L$. Let $\sigma \ge \frac{cL\sqrt{T\log(1/\delta)}}{n\epsilon}$ and $T \ge \frac{n^2\epsilon}{b^2}$. Then the algorithm which samples batches of size $B_1, .., B_t$ of size $b$ uniformly at random and outputs $\frac{1}{b}\sum_{y \in B_t} h_t(y) + g_t$ for all $t \in [T]$ where $g_t \sim \mathcal{N}(0, \mathbb{I}_d\sigma^2)$, is $(\epsilon, \delta)$-DP.*

### C.2  Proof of Lemma 3

Lemma 3 is easily established from the following theorem which holds more generally for any monotone operator (rather than just the saddle operator of a convex-concave function). This generalization will allow us to use this theorem again in our results on SVIs.

Recall we consider the norm $\|[w, \theta]\| = \frac{1}{B_w^2}\|w\|_p^2 + \frac{1}{B_\theta^2}\|\theta\|_q^2$ and have defined $\kappa = \max\left\{\frac{1}{\bar{p}-1}, \frac{1}{\bar{q}-1}\right\}$ and $\tilde{\kappa} = 1 + \mathbf{1}\{p < 2 \vee q < 2\} \cdot \log(d)$. For any $t \in \{0, ..., T\}$ define $[w_t, \theta_t] = z_t$, where $z_t$ is as given in Algorithm 2. We have the following.

**Theorem 4.** *Let $[w_0, \theta_0], [w, \theta]$ satisfy $\mathbb{E}\left[\|[w_0, \theta_0] - [w, \theta]\|\right] \le \hat{D}$. Let $g : \mathcal{W} \times \Theta \times \mathcal{X} \mapsto \mathcal{B}_{\|\cdot\|_{\bar{p}}}^{d_w}(B_wL_w) \times \mathcal{B}_{\|\cdot\|_{\bar{q}}}^{d_\theta}(B_\theta L_\theta)$ be a monotone operator and $G_S(w, \theta) = \frac{1}{n}\sum_{x \in S} g([w, \theta]; x)$. There exists an implementation of $\mathcal{O}$ such that Algorithm 2 is $(\epsilon, \delta)$-DP and the following holds*

$$\mathbb{E}\left[\langle G_S([w_{t^*}, \theta_{t^*}]), [w_{t^*}, \theta_{t^*}] - [w, \theta]\rangle\right] = \mathbb{E}\left[\frac{1}{T}\sum_{t=1}^{T}\langle G_S([w_t, \theta_t]), [w_t, \theta_t] - [w, \theta]\rangle\right]$$

$$= O\left(\hat{D}\sqrt{B_w^2L_w^2 + B_\theta^2L_\theta^2}\left(\frac{\sqrt{\kappa}\sqrt{d\log(1/\delta)\tilde{\kappa}}}{n\epsilon} + \frac{\sqrt{\kappa}}{\sqrt{n}}\right)\right)$$

*Further, the resulting algorithm makes* $O\left(\min\left\{\frac{\sqrt{\kappa}n^2\epsilon^{1.5}}{\log^2(n)\sqrt{d\log(1/\delta)\tilde{\kappa}}}, \frac{\sqrt{\kappa}n^{3/2}}{\log^{3/2}(n)}\right\}\right)$ *gradient evaluations.*

Before proving this statement, we first quickly show how to obtain Lemma 3, the relative accuracy guarantee for SSPs, using this result.

*Proof of Lemma 3.* To obtain Lemma 3 from this statement, observe that in the special case where $g$ is the saddle operator of the loss, $f$, convexity-concavity implies

$$\mathbb{E}\left[F(\frac{1}{T}\sum_{t=1}^{T}w_t, w) - F(\theta, \frac{1}{T}\sum_{t=1}^{T}\theta_t)\right] \leq \mathbb{E}\left[\frac{1}{T}\sum_{t=1}^{T}\langle G_{\mathcal{D}}(z_t), z_t - z\rangle\right],$$

and Lemma 3 is thus obtained from the bound in Theorem 4. □

All that remains is to prove the above theorem. Note the following proof leverages the structure $\mathcal{Z} = \mathcal{W} \times \Theta$, but does *not* assume that $g$ is the saddle operator of some convex-concave loss.

*Proof of Theorem 4.* Let $L = B_2^2 L_w^2 + B_\theta^2 L_\theta^2$ and let $p^* = \frac{\bar{p}}{\bar{p}-1}$ and $q^* = \frac{\bar{q}}{\bar{q}-1}$ be the conjugate exponents of $\bar{p}$ and $\bar{q}$, respectively. For $t \in [T]$ denote the result of $\mathcal{O}(z_t)$ as $[\nabla_{w,t}, \nabla_{\theta,t}]$ such that $\nabla_{w,t} \in \mathbb{R}^{d_w}$ and $\nabla_{\theta,t} \in \mathbb{R}^{d_\theta}$.

We consider the following construction of the private operator oracle, $\mathcal{O}$, for $G_S$. Our implementation adds Gaussian noise to minibatch estimates of $G_S$. That is, to evaluate $\mathcal{O}(z_t)$, we uniformly sample a minibatch of size $m = \max\left\{n\sqrt{\frac{\epsilon}{T}}, 1\right\}$, call it $M_t$, as well as Gaussian noise vectors $\xi_{w,t} \sim \mathcal{N}(0, \mathbb{I}_{d_w}\sigma_w^2)$ and $\xi_{\theta,t} \sim \mathcal{N}(0, \mathbb{I}_{d_\theta}\sigma_\theta^2)$, for some $\sigma_w, \sigma_\theta > 0$. We then have that

$$[\nabla_{w,t}, \nabla_{\theta,t}] = \frac{1}{m}\sum_{x \in M_t} g(w_t, \theta_t; x) + [\xi_{w,t}, \xi_{\theta,t}].$$

*Privacy Bound:* We first bound the privacy of Algorithm 2. Since for any $u$, $\|u\|_2 \leq \sqrt{d^{1-2/p^*}}\|u\|_{p*}$, we can bound the privacy loss using the guarantees of the moments accountant, Lemma 9. Specifically, we set $T = \kappa\min\left\{n, \frac{n^2\epsilon^2}{d\log(1/\delta)\tilde{\kappa}}\right\}$, $\eta = \frac{\hat{D}}{L\sqrt{T}}$, use minibatches of size $m = \max\left\{n\sqrt{\frac{\epsilon}{T}}, 1\right\}$, and set $\sigma_w = \frac{cB_w L_w\sqrt{Td_w^{1-2/p^*}\log(1/\delta)}}{n\epsilon}$ and $\sigma_\theta = \frac{cB_\theta L_\theta\sqrt{Td_\theta^{1-2/p^*}\log(1/\delta)}}{n\epsilon}$ for some universal constant $c$. It can be verified this scale of noise satisfies the conditions of Lemma 9 and thus ensures $(\epsilon, \delta)$-DP.

*Utility Bound:* We now establish the convergence guarantee by applying the general convergence guarantee of stochastic mirror prox (Lemma 8, Appendix C.1) with $\psi([w,\theta]) = \frac{\kappa}{2B_w^2}\|w\|_{\bar{p}}^2 + \frac{\kappa}{2B_\theta^2}\|\theta\|_{\bar{q}}^2$, which is 1-strongly convex w.r.t. $\|\cdot\|$. Clearly our saddle operator oracle yields unbiased estimates of $G_S(z)$ at each iteration. To bound the variance, $\tau$, note when $p \neq 2$ the private estimate of $\nabla_{w,t}$ satisfies

$$\mathbb{E}\left[\|\nabla_{w,t} - \nabla_w F_S(w_t, \theta_t)\|_{p^*}^2\right] \overset{(i)}{\leq} d_w^{2/p^*}\mathbb{E}\left[\|\xi_{w,t}\|_\infty^2\right] \leq d_w^{2/p^*}\sigma_w^2\log(d) \leq \frac{c^2B_w^2L_w^2Td_w\log(1/\delta)\log(d)}{n^2\epsilon^2}.$$

When $p = 2$, then $p^* = 2$, and one can replace bound $(i)$ with $\mathbb{E}\left[\|\xi_{w,t}\|_2^2\right]$. Combining these cases yields a bound of $\left(\frac{c^2B_w^2L_w^2Td_w\log(1/\delta)\tilde{\kappa}}{n^2\epsilon^2}\right)$ on the variance of $\nabla_{w,t}$. A similar analysis holds for $\nabla_{\theta,t}$. Ultimately, with respect to the norm chosen above, we get $\mathbb{E}\left[\|\mathcal{O}(z_t) - G(z_t)\|_*^2\right] \leq \tau^2$ with

$$\tau^2 \leq (B_w^2L_w^2 + B_\theta^2L_\theta^2)\left(1 + \frac{c^2Td\log(1/\delta)\tilde{\kappa}}{n^2\epsilon^2}\right) = O(L^2\kappa).$$

The last equality uses the fact that $\kappa \geq 1$. Recall we choose $\psi([w,\theta]) = \frac{\kappa}{2B_w^2}\|w\|_{\bar{p}}^2 + \frac{\kappa}{2B_\theta^2}\|\theta\|_{\bar{q}}^2$, and thus $\psi([w,\theta]) = \kappa\|[w,\theta]\|^2$. Now the guarantees of Lemma 8 imply

$$\mathbb{E}\left[\langle G_S(z_{t^*}), z_{t^*} - z\rangle\right] = \mathbb{E}\left[\frac{1}{T}\sum_{t=1}^{T}\langle G_{\mathcal{D}}(z_t), z_t - z\rangle\right] = O\left(\frac{\kappa\hat{D}^2}{T\eta} + \frac{7\eta}{2}(L^2 + 2\tau^2)\right).$$

The theorem statement now follows from plugging in the parameter settings of $T$ and $\eta$ and the bound on $\tau$ established above. □

# D Missing Results from Section 5

## D.1 Proof of Lemma 4

Let $h_1(z; x) = \langle g_1(z; x), z - z^* \rangle$. First, we observe that $h_1$ is $(\beta B + L)$-Lipschitz. To see this, we have for any $z, z' \in \mathcal{Z}$ and $x \in$ that

$$
\begin{aligned}
\langle g_1(z), z - z^* \rangle - \langle g_1(z'), z' - z^* \rangle &= \langle g_1(z) - g_1(z') + g_1(z'), z - z^* \rangle - \langle g_1(z'), z' - z^* \rangle \\
&= \langle g_1(z) - g_1(z'), z - z^* \rangle + \langle g_1(z'), z - z' \rangle \\
&\leq (\beta B + L) \|z - z'\|
\end{aligned}
$$

The rest of the proof is similar to existing stability implies generalization proofs, but with the additional accounting of the regularization term. In more detail, for any $i \in [n]$ denote $S^{(i)}$ as the dataset which replaces the $i$'th datapoint of $S$, $x_i$, with a fresh sample from $\mathcal{D}$, $x'$. We have the following:

$$
\underset{S, \mathcal{A}}{\mathbb{E}} [H_\mathcal{D}(\mathcal{A}(S)) - H_S(\mathcal{A}(S))]
$$

$$
= \underset{S, \mathcal{A}}{\mathbb{E}} \left[ \left\langle \underset{x}{\mathbb{E}} [g_1(\mathcal{A}(S); x)] + g_2(z), \mathcal{A}(S) - z^* \right\rangle - \left\langle \frac{1}{n} \sum_{x \in S} [g_1(\mathcal{A}(S); x)] + g_2(z), \mathcal{A}(S) - z^* \right\rangle \right]
$$

$$
= \underset{S, \mathcal{A}}{\mathbb{E}} \left[ \left\langle \underset{x}{\mathbb{E}} [g_1(\mathcal{A}(S); x)], \mathcal{A}(S) - z^* \right\rangle - \left\langle \frac{1}{n} \sum_{x \in S} [g_1(\mathcal{A}(S); x)], \mathcal{A}(S) - z^* \right\rangle \right]
$$

$$
= \underset{S, x' \sim \mathcal{D}^{n+1}, i \sim \mathsf{Unif}([n])}{\mathbb{E}} \left[ \left\langle g_1(\mathcal{A}(S^{(i)}); x_i), \mathcal{A}(S^{(i)}) - z^* \right\rangle - \langle g_1(\mathcal{A}(S); x_i), \mathcal{A}(S) - z^* \rangle \right]
$$

$$
= \mathbb{E} \left[ h_1(\mathcal{A}(S^{(i)}); x_i) - h_1(\mathcal{A}(S); x_i) \right]
$$

$$
\leq \mathbb{E} \left[ (\beta B + L) \|\mathcal{A}(S^{(i)}) - \mathcal{A}(S)\| \right] \leq (\beta B + L) \Delta.
$$

The last step follows from the previously established Lipschitzness property of $h_1$.

## D.2 Convergence of Recursive Regularization for SVIs

In this section, we define $\tilde{L} = \beta B + L$. Define $\mathrm{Gap}_{\mathsf{VI}}(z) = \max_{z'} \{\langle z', z - z' \rangle\}$. We have the following fact.

**Fact 5.** *If $g$ is $L$-bounded then $\widehat{\mathrm{Gap}_{\mathsf{VI}}}$ is $L$-Lipschitz.*

*Proof of 5.* For any $z_1, z_2 \in \mathcal{Z}$ we have

$$
\begin{aligned}
\widehat{\mathrm{Gap}_{\mathsf{VI}}}(z_1) - \widehat{\mathrm{Gap}_{\mathsf{VI}}}(z_2) &= \max_z \{\langle G_\mathcal{D}(z), z_1 - z \rangle\} - \max_{z'} \{\langle G_\mathcal{D}(z'), z_2 - z' \rangle\} \\
&\leq \max_z \{\langle G_\mathcal{D}(z), z_1 - z \rangle - \langle G_\mathcal{D}(z), z_1 - z \rangle\} \\
&= \max_z \{\langle G_\mathcal{D}(z), z_1 - z_2 \rangle\} \\
&\leq \|G_\mathcal{D}(z)\|_* \|z_1 - z_2\| \leq L \|z_1 - z_2\|.
\end{aligned}
$$

$\square$

We recall the assumption made on $\rho$.

**Assumption 2.** *For some $\kappa > 0$ let $\rho : \mathcal{Z} \mapsto \mathbb{R}^d$ be $\frac{1}{\kappa}$-strongly monotone w.r.t. $\| \cdot \|$ and satisfy $\|\rho(z)\|_* \leq \|z\|$ for all $z \in \mathcal{Z}$.*

We will first prove the following more general version statement of Theorem 2, which will be useful later.

**Theorem 5.** *Let $\lambda \geq \frac{48 \tilde{L} \kappa^2}{B \sqrt{n'}}$ and $\mathcal{A}_{\mathsf{emp}}$ be such that for all $t \in [T]$ it holds that $\mathbb{E} \left[ \|\bar{z}_t - z_{S,t}^*\|^2 \right] \leq \frac{B^2}{12 \cdot 2^{2t} \kappa^2}$. Then Recursive Regularization satisfies*

$$
\mathrm{Gap}_{\mathsf{VI}}(\mathcal{R}_{VI}) = O\left( \log(n) B^2 \lambda \right).
$$

To prove this result, it will be helpful to first show several intermediate results. We start by defining several useful quantities. Define $\{\mathcal{F}_t\}_{t=0}^T$ as the filtration where $\mathcal{F}_t$ is the sigma algebra induced by all randomness up to $\bar{z}_t$. For notational convenience we define $g^{(0)}(z; x) = g(z; x)$. Then for every $t \in \{0, 1, ..., T\}$ we define

- $z_t^*$ : equilibrium of $G_{\mathcal{D}}^{(t)}(z) := \underset{x \sim \mathcal{D}}{\mathbb{E}} \left[ g^{(t)}(z; x) \right]$;

- $z_{S,t}^*$ : equilibrium of $G_S^{(t)}(z) := \frac{1}{n'} \sum_{x \in S_t} g^{(t)}(z; x)$;

- $H_{\mathcal{D}}^{(t)}(\bar{z}) := \left\langle G_{\mathcal{D}}^{(t)}(\bar{z}), \bar{z} - z_t^* \right\rangle$ : the relative stationarity function w.r.t. $G_{\mathcal{D}}^{(t)}$ and $z_t^*$; and,

- $H_S^{(t)}(\bar{z}) := \left\langle G_S^{(t)}(\bar{z}), \bar{z} - z_t^* \right\rangle$ : the relative stationarity function w.r.t. $G_S^{(t)}$ and $z_t^*$.

As discussed in Section 5.2, the power of $H_{\mathcal{D}}^{(t)}$ is that it bounds the distance of a point to $z_t^*$.

**Fact 6.** *Let $G : \mathcal{Z} \mapsto \mathbb{R}^d$ be a $\mu$-strongly monotone operator and let $z^*$ be the equilibrium point. Then $\|z - z^*\|^2 \leq \frac{2\langle G(z), z - z^* \rangle}{\mu}$.*

*Proof.* By strong monotonicity, for any $z \in \mathcal{Z}$,

$$\frac{\mu}{2}\|z - z^*\|^2 \leq \langle G(z) - G(z^*), z - z^* \rangle \leq \langle G(z), z - z^* \rangle.$$

The last step comes from the fact that $\langle -G(z^*), z - z^* \rangle = \langle G(z^*), z^* - z \rangle \leq 0$ since $z^*$ is the equilibrium. $\qquad \square$

We now establish two distance inequalities which will be used when analyzing the final gap bound in Theorem 5. The first inequality below bounds the distance of the output of the $t$-th round to the equilibrium of $G_{\mathcal{D}}^{(t)}$. The second inequality bounds how far the population equilibria moves after another regularization term is added.

**Lemma 10.** *Assume the conditions of Theorem 5 hold. Then for every $t \in [T]$, the following holds*

**P.1** $\mathbb{E}\left[\|\bar{z}_t - z_t^*\|\right]^2 \leq \mathbb{E}\left[\|\bar{z}_t - z_t^*\|^2\right] \leq \frac{B^2}{2^{2t}\kappa^2}$; *and,*

**P.2** $B_t^2 := \mathbb{E}\left[\|z_t^* - z_{t-1}^*\|\right]^2 \leq \mathbb{E}\left[\|z_t^* - z_{t-1}^*\|^2\right] \leq \frac{B^2}{2^{2(t-1)}}$.

*Proof.* We will prove both properties via induction on $B_1, ..., B_T$. Specifically, for each $t \in [T]$ we will introduce two terms $E_t$ and $F_t$, and show that these terms are bounded if the bound on $B_t$ holds and that $B_t$ holds if $E_{t-1}$ and $F_{t-1}$ are bounded. Property **P.1** is then established as a result of the fact that $\mathbb{E}\left[\|\bar{z}_t - z_t^*\|^2\right] \leq 2(E_t + F_t)$. Note that $B_1$ holds as the base case because $\mathbb{E}\left[\|z_1^* - z_0^*\|^2\right] \leq B^2$.

**Property P.1:** We here prove that if $B_t$ is sufficiently bounded, then $E_t$ and $F_t$ are bounded where for $t \in [T]$ we define

$$E_t = \mathbb{E}\left[\|\bar{z}_t - z_{S,t}^*\|^2\right], \qquad\qquad F_t = \frac{\kappa}{2^t \lambda}\mathbb{E}\left[H_{\mathcal{D}}^{(t)}\left(z_{S,t}^*\right)\right]. \qquad (16)$$

Additionally, this will establish property **P.1** because for any $t \in [T]$ it holds that,

$$\mathbb{E}\left[\|\bar{z}_t - z_t^*\|^2\right] \leq 2\left(\mathbb{E}\left[\|\bar{z}_t - z_{S,t}^*\|^2\right] + \mathbb{E}\left[\|z_{S,t}^* - z_t^*\|^2\right]\right)$$

$$\leq 2\left(\underbrace{\mathbb{E}\left[\|\bar{z}_t - z_{S,t}^*\|^2\right]}_{E_t} + \underbrace{\frac{\kappa}{2^t \lambda}\mathbb{E}\left[H_{\mathcal{D}}^{(t)}\left(z_{S,t}^*\right)\right]}_{F_t}\right). \qquad (17)$$

The second inequality comes from the strong monotonicity of the operator (see Fact 6).

Since $E_t$ is bounded by the assumption made in the statement of Theorem 5, we focus on bounding $F_t$. We have

$$
\begin{aligned}
\frac{\kappa}{2^t\lambda}\mathbb{E}\left[H_{\mathcal{D}}^{(t)}\left(z_{S,t}^*\right)\right] &= \frac{\kappa}{2^t\lambda}\,\mathbb{E}\left[\mathbb{E}\left[H_{\mathcal{D}}^{(t)}\left(z_{S,t}^*\right)\Big|\mathcal{F}_{t-1}\right]\right] \\
&\leq \frac{\kappa}{2^t\lambda}\left(\mathbb{E}\left[\mathbb{E}\left[H_S^{(t)}\left(z_{S,t}^*\right)\Big|\mathcal{F}_{t-1}\right]\right]+\frac{\kappa\tilde{L}^2}{2^t\lambda n'}\right) \\
&= \frac{\kappa}{2^t\lambda}\left(\mathbb{E}\left[\mathbb{E}\left[\left\langle G_S^{(t)}(z_{S,t}^*),z_{S,t}^*-z_t^*\right\rangle\Big|\mathcal{F}_{t-1}\right]\right]+\frac{\kappa\tilde{L}^2}{2^t\lambda n'}\right) \\
&\leq \frac{\kappa^2\tilde{L}^2}{2^{2t}\lambda^2 n'}\leq \frac{B^2}{2304\cdot 2^{2t}\kappa^2}.
\end{aligned}
$$

The first inequality comes from the fact that stability implies generalization for $H^{(t)}$, Lemma 4. Note the algorithm which outputs this exact equilibrium point is $\frac{L}{2^t\lambda n'}$ uniform argument stable (see Lemma 5/Assumption 1). The second inequality comes from the fact that $z_{S,t}^*$ is the exact empirical equilibrium point of the regularized objective, and so for any $z\in\mathcal{Z}$, $\left\langle G_S^{(t)}(z_{S,t}^*),z_{S,t}^*-z\right\rangle\leq 0$. The final inequality uses the setting of $\lambda$.

We thus have a final bound $2(E_t+F_t)\leq\frac{B^2}{2^{2t}}$.

**Property P.2:** Now assume $B_{t-1}$ holds. We have

$$
\begin{aligned}
\mathbb{E}\left[\left\|z_t^*-z_{t-1}^*\right\|^2\right] &\leq \mathbb{E}\left[\frac{\kappa}{2^t\lambda}\left\langle G_{\mathcal{D}}^{(t)}(z_{t-1}^*),z_{t-1}^*-z_t^*\right\rangle\right] \\
&= \mathbb{E}\left[\frac{\kappa}{2^t\lambda}\left\langle G_{\mathcal{D}}^{(t-1)}(z_{t-1}^*)+2^t\lambda\rho(z_{t-1}^*-\bar{z}_{t-1}),z_{t-1}^*-z_t^*\right\rangle\right] \\
&= \mathbb{E}\left[\frac{\kappa}{2^t\lambda}\left\langle G_{\mathcal{D}}^{(t-1)}(z_{t-1}^*),z_{t-1}^*-z_t^*\right\rangle+\kappa\left\langle\rho(z_{t-1}^*-\bar{z}_{t-1}),z_{t-1}^*-z_t^*\right\rangle\right] \\
&\overset{(i)}{\leq}\mathbb{E}\left[\frac{\kappa^2}{2}\left\|\rho(z_{t-1}^*-\bar{z}_{t-1})\right\|_*^2+\frac{1}{2}\left\|z_{t-1}^*-z_t^*\right\|^2\right] \\
&\overset{(ii)}{\leq}\mathbb{E}\left[\frac{\kappa^2}{2}\left\|z_{t-1}^*-\bar{z}_{t-1}\right\|^2+\frac{1}{2}\left\|z_{t-1}^*-z_t^*\right\|^2\right].
\end{aligned}
$$

Inequality $(i)$ above comes from Young's inequality and the fact that $z_{t-1}^*$ is the equilibrium point w.r.t. $G_{\mathcal{D}}^{(t-1)}$. Inequality $(ii)$ comes from Fact 2/Assumption 1. After re-arranging we can continue as follows:

$$
\mathbb{E}\left[\left\|z_t^*-z_{t-1}^*\right\|^2\right]\leq\kappa^2\mathbb{E}\left[\left\|z_{t-1}^*-\bar{z}_{t-1}\right\|^2\right]\leq\kappa^2(E_{t-1}+F_{t-1})\leq\frac{B^2}{2^{2t}}.
$$

$\square$

We now turn to analyzing the utility of the algorithm to complete the proof.

*Proof of Theorem 5.* Using the fact that $\widehat{\mathrm{Gap}_{\mathsf{VI}}}$ is $L$-Lipschitz and property **P.1**, we have

$$
\begin{aligned}
\mathbb{E}\left[\widehat{\mathrm{Gap}_{\mathsf{VI}}}(\bar{z}_T)-\widehat{\mathrm{Gap}_{\mathsf{VI}}}(z_T^*)\right] &\leq L\mathbb{E}\left[\left\|\bar{z}_T-z_T^*\right\|\right] \\
&\leq\frac{BL}{2^T}\leq B^2\lambda. \quad\quad\quad (18)
\end{aligned}
$$

Note that because the above is a statement with respect to the unregularized gap function, we do not have to worry about whether or not the regularization term is smooth.

What remains is showing $\mathbb{E}\left[\widehat{\mathrm{Gap}_{\mathsf{VI}}}(z_T^*)\right]=O(\log(n)B^2\lambda)$. By the definition of $G_{\mathcal{D}}^{(T)}$ we have

$$
G_{\mathcal{D}}(z)=G_{\mathcal{D}}^{(T)}(z)-2\lambda\sum_{t=0}^{T-1}2^{t+1}\rho(z-\bar{z}_t)
$$

Let $z' = \arg\max_{z' \in \mathcal{Z}} \{\langle G_{\mathcal{D}}(z'), z_T^* - z'\rangle\}$. We obtain the following bound on the $\widehat{\mathrm{Gap_{VI}}}(w_T^*, \theta_T^*)$.

$$\widehat{\mathrm{Gap_{VI}}}(z_T^*) = \left\langle G_{\mathcal{D}}^{(T)}(z'), z_T^* - z'\right\rangle + \left\langle 2\lambda \sum_{t=0}^{T-1} 2^{t+1}\rho(z' - \bar{z}_t), z' - z_T^*\right\rangle$$

$$\overset{(i)}{\leq} \left\langle 2\lambda \sum_{t=0}^{T-1} 2^{t+1}\rho(z' - \bar{z}_t), z' - z_T^*\right\rangle$$

$$\overset{(ii)}{\leq} \left\langle 2\lambda \sum_{t=0}^{T-1} 2^{t+1}\rho(z_T^* - \bar{z}_t), z_T^* - z'\right\rangle$$

$$\overset{(iii)}{\leq} 2B\lambda \sum_{t=0}^{T-1} 2^{t+1}\|\rho(z_T^* - \bar{z}_t)\|_*$$

$$\overset{(iv)}{\leq} 2B\lambda \sum_{t=0}^{T-1} 2^{t+1}\|z_T^* - \bar{z}_t\|.$$

Above, $(i)$ comes from the fact that $z_T^*$ is the equilibrium point of $G_{\mathcal{D}}^{(T)}$. Inequality $(ii)$ uses monotonicity of $\rho$, i.e. $0 \leq \langle \rho(z_T^* - \bar{z}_T) - \rho(z' - \bar{z}_T), z_T^* - z'\rangle$. Inequality $(iii)$ comes from Holder's inequality and a triangle inequality. Finally, $(iv)$ comes from Assumption 1.

Taking the expectation on both sides of the above we have the following derivation,

$$\mathbb{E}\left[\widehat{\mathrm{Gap_{VI}}}(z_T^*)\right] \leq 2B\mathbb{E}\left[\lambda \sum_{t=0}^{T-1} 2^{t+1}\|z_T^* - \bar{z}_t\|\right]$$

$$\overset{(i)}{\leq} 4B\mathbb{E}\left[\lambda \sum_{t=0}^{T-1} 2^t \left(\|z_t^* - \bar{z}_t\| + \sum_{r=t}^{T-1}\|z_{r+1}^* - z_r^*\|\right)\right]$$

$$= 4B\mathbb{E}\left[\lambda \sum_{t=0}^{T-1} 2^t \|z_t^* - \bar{z}_t\| + \lambda \sum_{t=0}^{T-1} 2^t \sum_{r=t}^{T-1}\|z_{r+1}^* - z_r^*\|\right]$$

$$\overset{(ii)}{=} 4B\mathbb{E}\left[\lambda \sum_{t=0}^{T-1} 2^t \|z_t^* - \bar{z}_t\| + \lambda \sum_{r=0}^{T-1}\sum_{t=0}^{r-1} 2^t \|z_{r+1}^* - z_r^*\|\right]$$

$$= 4B\mathbb{E}\left[\lambda \sum_{t=0}^{T-1} 2^t \|z_t^* - \bar{z}_t\| + \lambda \sum_{r=0}^{T-1}\|z_{r+1}^* - z_r^*\|\sum_{t=0}^{r-1} 2^t\right]$$

$$\overset{(iii)}{\leq} 4B\left(\lambda \sum_{t=0}^{T-1} 2^t \left(\frac{B}{2^t}\right) + \lambda \sum_{r=1}^{T-1}\left(\frac{B}{2^r}\right)\sum_{t=0}^{r-1} 2^t\right)$$

$$\leq 4B\left(\lambda \sum_{t=0}^{T-1} 2^t \left(\frac{B}{2^t}\right) + \lambda \sum_{r=1}^{T-1}\left(\frac{B}{2^{r-1}}\right)\cdot(2^r - 1)\right)$$

$$= 4\lambda \sum_{t=0}^{T-1} B^2 + 8\lambda \sum_{r=1}^{T-1} B^2$$

$$\leq 12T\lambda B^2 \qquad (19)$$

Above, $(i)$ and the following inequality both come from the triangle inequality. Equality $(ii)$ is obtained by rearranging the sums. Inequality $(iii)$ comes from applying properties **P.1** and **P.2** proved above. The last equality comes from the setting of $\lambda$ and $T$.

Now using this result in conjunction with Eqn. (18) we have

$$\mathrm{Gap_{VI}}(\mathcal{R}_{\mathrm{SVI}}) = \sqrt{2}\lambda B^2 + 12T\lambda B^2 = O\left(\log(n)B^2\lambda\right).$$

Above we use the fact that $T = \log(\frac{L}{B\lambda})$ and $\lambda \geq \frac{L}{B\sqrt{n'}}$, and thus $T = O(\log(n))$. $\qquad\square$

Finally, we prove Theorem 2 leveraging the relative stationarity assumption.

*Proof of Theorem 2.* First, observe that under the setting of $\lambda = \frac{48}{B}\left(\hat{\alpha}\kappa^3 + \frac{\tilde{L}\kappa^2}{\sqrt{n'}}\right)$ used in the theorem statement that $\log(n)B^2\lambda = O\left(\log(n)B\hat{\alpha}\kappa^3 + \frac{\log^{3/2}(n)B\tilde{L}\kappa^2}{\sqrt{n}}\right)$. Thus what remains is to show that the distance condition required by Theorem 5 holds. That is, we now show that if $\mathcal{A}_{\mathsf{emp}}$ satisfies $\hat{\alpha}$-relative stationarity, then for all $t \in [T]$ it holds that $\mathbb{E}\left[\left\|\bar{z}_t - z_{S,t}^*\right\|^2\right] \leq \frac{B^2}{12 \cdot 2^{2t}\kappa^2}$.

To prove this property, we must leverage the induction argument made by Lemma 10. Specifically, to prove the condition holds for some $t \in [T]$, assume $B_t^2 = \mathbb{E}\left[\left\|z_t^* - z_{t-1}^*\right\|\right]^2 \leq \frac{B^2}{2^{2(t-1)}}$ (recall the base case for $t = 1$ trivially holds). As shown in the proof of Lemma 10, this implies that the quantities $E_t, F_t$ (as defined in 16) are bounded by $\frac{B^2}{2304 \cdot 2^{2t}}$. We thus have

$$\mathbb{E}\left[\left\|\bar{z}_t - z_{S,t}^*\right\|^2\right] \overset{(i)}{\leq} \frac{\kappa\mathbb{E}\left[\left\langle G_S^{(t)}(\bar{z}_t), \bar{z}_t - z_{S,t}^*\right\rangle\right]}{2^t\lambda} \overset{(ii)}{\leq} \frac{2\kappa\hat{\alpha}B}{2^{2t}\lambda} \overset{(iii)}{\leq} \frac{B^2}{12 \cdot 2^{2t}\kappa^2}, \qquad (20)$$

where $B_t$ is as defined in property **P.2**. Inequality $(i)$ comes from the strong monotonicity of $G_S^{(t)}$, Fact 6. Inequality $(iii)$ comes from the setting $\lambda \geq 48\hat{\alpha}\kappa/B$. Inequality $(ii)$ comes from the $\hat{\alpha}$-relative stationarity assumption on $\mathcal{A}_{\mathsf{emp}}$, which holds so long as the expected distance is sufficiently bounded and the operator is bounded. In this regard, note that

$$\mathbb{E}\left[\left\|z_{S,t}^* - \bar{z}_{t-1}\right\|\right] \leq \mathbb{E}\left[\left\|z_{S,t}^* - z_t^*\right\| + \left\|z_t^* - z_{t-1}^*\right\| + \left\|z_{t-1}^* - \bar{z}_{t-1}\right\|\right]$$

$$\leq (\sqrt{F_t} + B_t + \sqrt{E_{t-1}} + \sqrt{F_{t-1}}) \leq \frac{2B}{2^t}.$$

Further, each $g^{(t)}$ is $5L$-bounded. That is, observe

$$\max_{z \in \mathcal{Z}}\left\|g^{(t)}(z, x)\right\|_* \leq \left\|z\right\|_* + \sum_{k=0}^{t-1} 2^{k+1}\lambda\left\|\rho(z - \bar{z}_t)\right\|_* \leq L + \sum_{k=0}^{t-1} B2^{k+1}\lambda \leq L + 4B2^T\lambda \leq 5L.$$

$\square$

### D.3 Near Linear Time Algorithm for SVIs in the $\ell_2$ Setting

Because we assume the operator is Lipschitz, in the $\ell_2$ setting, we can leverage existing accelerated optimization techniques to achieve a near linear time version of $\mathcal{A}_{\mathsf{emp}}$, in a similar fashion to [ZTOH22, BGM23]. Specifically, the work [PB16] gives the following result for strongly monotone variational inequalities when applying their accelerated SVRG algorithm [3].

**Lemma 11.** *(Implicit in [PB16, Theorem 3]) Let $\beta, \mu, K > 0$ and $c$ a universal constant. Let $g : \mathcal{Z} \times \mathcal{X} \mapsto \mathbb{R}$ be monotone and $\beta$-Lipschitz and $\rho : \mathcal{Z} \mapsto \mathbb{R}$ a $\mu$-strongly monotone operator. Let $z_S^*$ be the equilibrium of $G_S(z) = \frac{1}{n}\sum_{x \in S} g(z; x) + \rho(z)$. There exists an algorithm, which in $O(n + \sqrt{n}K\frac{\beta}{\mu})$ gradient evaluations find a point $\bar{z}$ such that $\mathbb{E}\left[\left\|z^* - \bar{z}\right\|_2\right] = cBe^{-K}$.*

We now construct $\mathcal{A}_{\mathsf{emp}}$ in the following way. At each round $t \in [T]$, we use the accelerated algorithm mentioned above to find a point $\hat{z}$ such that $\mathbb{E}\left[\left\|\hat{z} - z_{S,t}^*\right\|_2\right] \leq \left(\frac{\delta L}{52^t\lambda n'}\right)$, where $z_{S,t}^*$ is the equilibrium point of $G_S^{(t)}(z) = \frac{1}{n'}\sum_{x \in S} g^{(t)}(z; x)$. We then have $\mathcal{A}_{\mathsf{emp}}$ output the point $\bar{z}_t = \hat{z}_t + \xi_t$, where $\xi_t \sim \mathcal{N}(0, \mathbb{I}_d\sigma_t^2)$ and $\sigma_t = \frac{8L\sqrt{2/\delta}}{2^t\lambda n'\epsilon}$. Using this construction, we can obtain the following result.

**Theorem 6.** *Let $\mathcal{A}_{\mathsf{emp}}$ be as described above. Then Algorithm 1 is $(\epsilon, \delta)$-DP and when run with $\lambda = \frac{48}{B}\left(\frac{L}{\sqrt{n'}} + \frac{L\sqrt{d\log(2/\delta)}}{n'\epsilon}\right)$ satisfies*

$$\mathrm{Gap}(\mathcal{R}_{SVI}) = O\left(\frac{\log^{3/2}(n)BL}{\sqrt{n}} + \frac{\log^2(n)BL\sqrt{d\log(1/\delta)}}{n\epsilon}\right),$$

*and runs in at most $O(n + \beta n\log(n/\delta))$ gradient evaluations.*

---

[3]In the main body of [PB16], the authors discuss only saddle point problems. However, they prove their result more generally for monotone operators. See their discussion in Section 6 and Appendix A of their paper.

*Proof.* *Privacy Guarantee* We show that $\mathcal{A}_{\mathsf{emp}}$ is $(\epsilon, \delta)$ at any iteration $t \in [T]$ using the stability properties of the regularized operator. Specifically, the exact equilibrium to $G_S^t$. Now because $\mathcal{A}_{\mathsf{emp}}$ guarantees $\mathbb{E}\left[\|\hat{z} - z_{S,t}^*\|_2\right] \leq \left(\frac{\delta L}{52^t \lambda n'}\right)$, an application of Markov's inequality implies that with probability at least $1 - \delta$ that $\|\hat{z}_t - z_{S,t}^*\|_2 = \left(\frac{1}{\delta}\left(cBe^{-K}\right)\right) \leq \frac{L}{2^T \lambda n}$. Thus by a triangle inequality $\hat{z}_t$ is $\left(\frac{2L}{2^t \lambda n'}\right)$-stable with probability at least $1 - \delta$. The guarantees of the Gaussian mechanism thus imply $\mathcal{A}_{\mathsf{emp}}$ is $(\epsilon, \delta)$-DP.

*Utility Guarantee:* We prove the utility guarantee by leveraging Theorem 5, which guarantees $\mathrm{Gap}_{\mathsf{VI}}(\mathcal{R}) = O(\log(n)B^2\lambda) = O\left(\frac{\log^{3/2}(n)BL}{\sqrt{n}} + \frac{\log^2(n)BL\sqrt{d\log(1/\delta)}}{n\epsilon}\right)$, so long as for every $t \in [T]$ it holds that $\mathbb{E}\left[\|\bar{z}_t - z_{S,t}^*\|_2^2\right] \leq \frac{B^2}{12 \cdot 2^{2t}}$. Note here that under the choice of $\rho$ in Eqn. (7), we satisfy Assuption 1 with $\kappa = 1$. We thus finish the proof with the following analysis,

$$\mathbb{E}\left[\|\bar{z}_t - z_{S,t}^*\|_2^2\right] = \mathbb{E}\left[\|\xi_t\|_2^2 + \|\hat{z}_t - z_{S,t}^*\|_2^2\right]$$

$$\leq d\sigma_t^2 + \left(\frac{\delta}{5} \cdot \frac{L}{2^t \lambda n'}\right)^2$$

$$\leq \frac{64dL^2 \log(2/\delta)}{2^{2t}\lambda^2(n')^2\epsilon^2} + \frac{B^2}{25 \cdot 2^{2t}} \leq \frac{B^2}{12 \cdot 2^{2t}}.$$

*Running Time:* By the guarantees of Lemma 11, we can achieve the condition $\mathbb{E}\left[\|\hat{z} - z_{S,t}^*\|_2\right] \leq \left(\frac{\delta L}{52^t \lambda n'}\right)$ made in the description of $\mathcal{A}_{\mathsf{emp}}$ by setting $K = \log\left(\frac{cB}{\delta} \cdot \frac{2^T \lambda n}{L}\right)$. Recall $T = \log_2\left(\frac{L}{\kappa B\lambda}\right) \leq n$. Thus the overall running time is $O(n + \frac{\beta}{\mu}K\sqrt{n}) = O(n + \beta n \log(n/\delta))$.

$\square$

# E   Lower Bound for SVIs

The lower bound for SVIs in the $\ell_2$ setting was established in [BG23]. Their technique can easily be extended to other geometries. Specifically, the lower bound comes from two observations. First, for linear losses, the strong VI-gap is equal to the excess population risk when the operator in question is the gradient. Second, the nearly tight lower bound constructions for DP stochastic minimization problems use linear losses.

We establish the first fact more formally here.

**Lemma 12.** *Let $f(z; x) = \langle z, x \rangle$ and define the operator $g(z; x) = \nabla f(z; x) = x$. Then $\mathrm{Gap}_{\mathsf{VI}}(z) = F_{\mathcal{D}}(z) - \min_{u \in \mathcal{Z}}\{F_{\mathcal{D}}(u)\}$. That is, the strong VI-gap w.r.t. $g$ is equal to the excess population risk w.r.t. the $f$.*

*Proof.* We have

$$\mathrm{Gap}_{\mathsf{VI}}(z) = \max_u \left\{\left\langle \mathbb{E}_{x \sim \mathcal{D}}[x], z - u \right\rangle\right\}$$

$$= \left\langle \mathbb{E}_{x \sim \mathcal{D}}[x], z \right\rangle + \max_u \left\{\left\langle \mathbb{E}_{x \sim \mathcal{D}}[x], -u \right\rangle\right\}$$

$$= F_{\mathcal{D}}(z) - \min_{u \in \mathcal{Z}}\{F_{\mathcal{D}}(u)\}.$$

$\square$

We now restate the lower bound result of [BGN21] for non-Euclidean setups.

**Theorem 7.** *([BGN21, Theorem 7.1]) Let $p \in (1, 2)$ and $p^* = \frac{p}{p-1}$ and $\mathcal{Z} = \mathcal{B}_{\|\cdot\|_p}(1)$. Let $\epsilon > 0$ and $0 < \delta < \frac{1}{n^{1+\Omega(1)}}$ and let $f(z; x) = \langle z, x \rangle$. For any $(\epsilon, \delta)$-DP algorithm $\mathcal{A}$, there exists a distribution $\mathcal{D}$ over $\mathcal{X} = \mathcal{B}_{\|\cdot\|_{p^*}}(1)$ such that*

$$\mathbb{E}_{S \sim \mathcal{D}^n, \mathcal{A}}\left[F\mathcal{D}(\mathcal{A}(S) - \min_{z \in \mathcal{Z}}\{F_{\mathcal{D}}(z)\}\right] = \tilde{\Omega}\left(\max\left\{\frac{1}{\sqrt{n}}, \frac{(p-1)\sqrt{d}}{n\epsilon}\right\}\right).$$

Note that a linear dependence on $BL$ can be obtained using classic rescaling arguments. The two above results (and the result of [BG23]) then imply that the strong VI-gap rate we obtain, $\tilde{O}\left(BL \max\left\{\frac{1}{\sqrt{n}}, \frac{(p-1)\sqrt{d}}{n\epsilon}\right\}\right)$, is near optimal for $p \in (1, 2]$ when $p - 1 = \Omega(1)$.

