# OpenReview forum: "Private Algorithms for Stochastic Saddle Points and Variational Inequalities: Beyond Euclidean Geometry"
_NeurIPS.cc/2024/Conference — NeurIPS 2024 poster_

### Official Review · Reviewer_nH8p · 2024-07-01

**Soundness:** 2
**Presentation:** 2
**Contribution:** 3
**Rating:** 5
**Confidence:** 2

**Summary:**

This paper studies private SSP beyond Euclidean geometry. They prove a near optimal bound on SP-gap for geometry between $\ell_1 ,\ell_2$. This result is then extended to SVI.

**Strengths:**

The results are solid improvement over previous work. The method on overcoming the generalization issue is novel and interesting.

**Weaknesses:**

I have a few concerns on the technical results.

(1) what’s $A_{emp}$ in your algorithm. It’s the key subroutine while never formally defined or specified. Can you give a concrete example as you claimed in line 182?

(2) is there any algorithmic novelty compared with previous work, or it’s just an improved analysis? The comparison with BGM23 can be made more clear.

(3) can you give some practical examples to prompt the need of considering noneuclidean geometry, otherwise it looks somewhat incremental.

There are numerous typos.

**Questions:**

See weaknesses.

**Limitations:**

See weaknesses.

---

> ### Author Rebuttal · Authors · 2024-08-03
>
> 1. The exact instantiation of $\mathcal A_{emp}$ we use is given by Algorithm 2, Stochastic Mirror Prox. Lemma 3 shows that the implementation satisfies the needed relative accuracy guarantee. We will add the following comment to the ``Algorithm Overview'' section, line 179, to make this more clear.
>
>
>    *The saddle point problem defined in each round of Algorithm 1 is solved using some empirical subroutine, $\mathcal A_{emp}$. This subroutine takes as input a partition of the dataset, $S_t$, the regularized loss function for that round, $f^{(t)}$, a starting point, $[\bar w_{t-1},\bar \theta_{t-1}]$, and an upper bound on the expected distance to the empirical saddle point of the problem defined by $S_t$ and $f^{(t)}$. The exact implementation of $\mathcal A_{emp}$, Algorithm 3, will be discussed in the next section. Here, we focus on the guarantees of Recursive Regularization given that $\mathcal A_{emp}$ satisfies a certain accuracy condition.*
>
>
> 2. There is some algorithmic novelty in that we 1) need to use non-Euclidean regularizes and 2) need to implement new DP techniques for the subroutine $\mathcal A_{emp}$. That is, for the purposes of [BGM23], noisy stochastic gradient descent-ascent was sufficient, but our more general setup required a private implementation of the stochastic mirror prox algorithm. With that said, our main claim to novelty is in our analysis, which differs in crucial and non-obvious ways from [BGM23], as we detail in Section 3.
>
> 3. Yes, the $\ell_1/\ell_2$ setup is particularly important. This is used to formulate problems that allow an adversary to mix different possible loss functions. Concretely, assume $f_1(w;x),...,f_k(w;x)$ are $\ell_2$-Lipschitz loss functions. Then one can consider the saddle point problem:
>
>     $$ F_{\mathcal D}(w,\theta) = \mathbb E_{x\sim\mathcal D}\Big[{\sum_{j=1}^k \theta_j f_j(w;x)}\Big],$$
>
>     where $\theta$ is constrained to the standard simplex and $w$ is constrained to some compact, $\ell_2$-bounded set.
>     This setup has been used in agnostic federated learning [MSS19] as just one example. We will add this example to our paper using the extra space given for the revision.
>
>     [MSS19]: Mehryar Mohri, Gary Sivek, and Ananda Theertha Suresh. Agnostic federated learning. ICML 2019

---

> > ### Comment · Reviewer_nH8p · 2024-08-08
> >
> > Thank you for your response! The paper can benefit from adding these explanations/discussions. I will maintain my score.

---

### Official Review · Reviewer_sj99 · 2024-07-17

**Soundness:** 3
**Presentation:** 3
**Contribution:** 3
**Rating:** 5
**Confidence:** 2

**Summary:**

This paper is quite far from my area, so please consider this review accordingly.

The paper addresses the problem of private Stochastic Saddle Points and Variational Inequalities. The primary contribution is extending previous work that focused solely on the L2/L2 setup to more general lp/lq settings, where the primal problem follows an lp-setup and the dual problem follows an lq-setup.

The main result is the development of an algorithm that achieves optimal excess error measured by the strong SP-gap.

**Strengths:**

The paper, in general, feels quite dense. For instance, the second paragraph mentions monotone operators without providing a definition. Additionally, the contribution section is not clear to me.

The algorithmic aspect of the work is very similar to [BGM23]. However, the authors needed to make some changes to the analysis, and they did a good job describing these necessary modifications.

**Weaknesses:**

The main weakness of the work is its presentation. It is very difficult to parse many parts of the paper.

**Questions:**

n/a

**Limitations:**

yes

---

> ### Author Rebuttal · Authors · 2024-08-03
>
> We provide a definition of monotone operators in the preliminaries section, line 122. We can use the extra page allowed in the final version to provide more background.
>
> With regards to the contribution of our work, while some algorithmic changes are needed in comparison to [BGM23], we emphasize that our primary contribution is our analysis technique.
> There is some algorithmic novelty in that we 1) need to use non-Euclidean regularizes and 2) need to implement new DP techniques for the subroutine $\mathcal{A}_{emp}$. That is, for the purposes of [BGM23], noisy stochastic gradient descent-ascent was sufficient, but our more general setup required a private implementation of the stochastic mirror prox algorithm. With that said, our main claim to novelty is in our analysis, which differs in crucial and non-obvious ways from [BGM23], as we detail in Section 3.

---

### Official Review · Reviewer_9vNb · 2024-07-20

**Soundness:** 3
**Presentation:** 3
**Contribution:** 2
**Rating:** 5
**Confidence:** 4

**Summary:**

This work studied stochastic saddle point and variational inequality problems in potentially non-Euclidean cases.  For stochastic saddle point problems, they proposed a recursive regularization framework, and provided the convergence guarantee and sample complexity for convex-concave problems. They further extended the framework to variational inequalities and incorporated differential privacy. Corresponding convergence guarantees and complexities results are also provided.

**Strengths:**

1. First work on SSPs and SVIs in general non-Euclidean settings.
2. The proposed rate is nearly optimal

**Weaknesses:**

1. The boundedness assumption is a bit restricted, regarding many unconstrained problems in practice.
2. Some important assumptions are hidden in the statement of Theorems, for example, the strong convexity of $||\cdot||_\omega$  and $||\cdot||_o$, while it is not fully rationalized (beyond $\ell_p, p\in(1,2]$ case), and it may not be satisfied in some important special case like $\ell_1$, I think the motivation for non-Euclidean
3. The paper flow and main results are a bit similar to [BGM23], which makes it a little incremental. Even though the authors claimed some differences, but from the appendix, many proofs are still very similar to those in [BGM23] with minor changes like $\kappa$. But I agree the changes in some parts like the proof of Property P.2 reveal certain novelty.

Typo:
1. In Algorithms, the parameters of $\mathcal{A}_{\text{emp}}(\cdot,\cdot,\cdot,\cdot)$ are not clearly defined

**Questions:**

/

**Limitations:**

/

---

> ### Author Rebuttal · Authors · 2024-08-03
>
> 1. Assuming the parameter space is bounded is very common in SSPs due to the problems unconstrained domains incur. For example, even for simple bilinear losses, say $f(w,\theta) = \langle w, \theta \rangle$, an unbounded domain means the strong gap is *infinite* at any non-zero point.
>
> 2. It is indeed not always the case that $\lVert \centerdot \rVert_w^2$ and $\lVert \centerdot \rVert_\theta^2$ are strongly convex. However, as we elaborate more in Section 4, this assumption is satisfied for $p\in[1+\frac{1}{\log(d)}, 2]$, where the squared norm is *strongly convex*. Further, for $p=1$ (and more generally, for $p\in[1,1+\frac{1}{\log(d)}]$), we easily solve the problem by instead solving a problem with $p'=1+\frac{1}{\log(d)}$, as we describe in Section 4. We will add more discussion after Theorem 1 so that this point does not feel hidden from the reader.
>
> 3. While [BGM23] contains several ideas that serve as a starting point for the current submission, note that this work makes key novel contributions which allow the nontrivial extensions to SVIs and non-Euclidean settings, as acknowledged by reviewer nH8p. Most importantly, our new generalization analysis for these problems (see `key proof ideas' in page 6) permits the use of sequential regularization in SVIs and non-Euclidean settings. To our knowledge, this result is entirely new, and of interest beyond differential privacy.
>
> 4. We will clarify the parameters of $A_{emp}$ by adding the following comment to the ``Algorithm Overview'' section, line 179:
>
> *The saddle point problem defined in each round of Algorithm 1 is solved using some empirical subroutine, $A_{emp}$. This subroutine takes as input a partition of the dataset, $S_t$, the regularized loss function for that round, $f^{(t)}$, a starting point, $[\bar w_{t-1}, \bar \theta_{t-1}]$, and an upper bound on the expected distance to the empirical saddle point of the problem defined by $S_t$ and $f^{(t)}$. The exact implementation of $A_{emp}$, Algorithm 3, will be discussed in the next section. Here, we focus on the guarantees of Recursive Regularization given that $A_{emp}$ satisfies a certain accuracy condition.*

---

### Official Review · Reviewer_mPfo · 2024-07-30

**Soundness:** 3
**Presentation:** 3
**Contribution:** 3
**Rating:** 6
**Confidence:** 2

**Summary:**

The authors study differentially private algorithms for stochastic saddle point (SSP) problems and stochastic variational inequalities (SVI). The proposed method relies on recursive regularization approach and obtain near optimal rates for settings were the parameters of interest are constrained to be in a bounded \ell_p ball and p \in [1,2].

**Strengths:**

The methods proposed by the authors recover many existing optimal results using different proof techniques. They extend the scope of the existing results with a unified analysis.

**Weaknesses:**

The paper does not seem to be self-contained. Some important components of the algorithms are not explicitly described, which makes it difficult to verify some of the claims in this work. The authors should include more details in the supplementary materials.

**Questions:**

I have two main comments:

1. The subroutine $\mathcal{A}_{emp}$ is never explicitly introduced. Here are some of sources of confusion for the reader:
-  Is this subroutine the same one in Algorithms 1 and 3?
-  In line 6 of Algorithm 1 the subroutine takes 4 inputs, none of which seems to be related to privacy parameters ?. However,  in lines 258-258 the authors say that the privacy of Algorithm 1 follows from the privacy of $\mathcal{A}_{emp}$.
- The construction of $\mathcal{A}_{emp}$ in lines 754-757 consists of taking the output of the SVRG algorithm of Palaniappan and Bach (2016) and add Gaussian noise to it? I think the precise SVRG algorithm the authors have in mind should also be presented. Some none trivial adaptations seems to be required.

2. Algorithm 1 requires $T=O(\log n)$ since $\lambda\geq \frac{K\kappa}{B\sqrt{n}}$ and $T=\log_2(\frac{L}{B\lambda})$. However, Lemma 3 requires a much larger number of gradient evaluations of roughly $\tilde{\Omega}(n^{3/2})$ to get the claimed accuracy that in turn is also needed in Corollary 1? Essentially the same comment applies to Theorem 2 and Corollary 2.

**Limitations:**

The nature of this work is theoretical but it is natural to wonder if the methods are easy to implement. Have the authors tried to run any numerical experiments?

---

> ### Author Rebuttal · Authors · 2024-08-03
>
> 1. We will add the following text to the ``Algorithm Overview'' paragraph (line 179), as well as additional comments:
>
> *The saddle point problem defined in each round of Algorithm 1 is solved using some empirical subroutine, $A_{emp}$. This subroutine takes as input a subset of the dataset, $S_t$, the regularized loss function for that round, $f^{(t)}$, a starting point, $[\bar w_{t-1}, \bar \theta_{t-1}]$, and an upper bound on the expected distance to the empirical saddle point of the problem defined by $S_t$ and $f^{(t)}$. The exact implementation of $A_{emp}$, Algorithm 3, will be discussed in the next section. Here, we focus on the guarantees of Recursive Regularization given that $A_{emp}$ satisfies a certain accuracy condition.*
>
> Additionally to your other points:
>
> - Yes, we state in the paper that we use Algorithm 3 for the subroutine when discussing both SSPs and SVIs; see lines 241-244 and 321.
> - The subroutine does indeed depend on the privacy parameters when one is interested in implementing Recursive Regularization in a private way. However 1) there may be value in our algorithm/analysis beyond privacy and 2) the privacy parameters do not change for each run of $\mathcal{A}_{emp}$, and so omitting them reduces cumbersome notation.
> - We can include the pseudocode for SVRG if the reviewer feels it is necessary, but we are unaware of the non-trivial changes being referred to, as we use it as a black box. Note that [PB16] details their algorithm for monotone operators in their appendix, as we state in our paper; see footnote 2 on page 24.
>
> 2. Algorithm 1 runs in roughly $T=O(\log n)$ rounds. Each round uses runs the subroutine $\mathcal{A}_{emp}$. To obtain Corollary 1, we show an implementation of this subroutine which runs in roughly $n^{3/2}$ gradient evaluations. Thus the overall running time of the algorithm is $\tilde{O}({n^{3/2}})$ gradient evaluations.

---

### Decision · Program_Chairs · 2024-09-25

**Decision:**

Accept (poster)

**Comment:**

The paper considers differentially private algorithms for stochastic saddle point and variational inequality problems. Its key contribution is that the proposed methods can deal efficiently with non-Euclidean norms, obtaining near-optimal rates. This requires an analysis of regularizations of the function under consideration, which leads to new insights.

The general evaluation of the reviewers was that this is a borderline/weak accept paper (scores were 5,5,5,6). It seems that some of the perceived weaknesses, such as the lack of adequate definition of $\mathcal{A}_{\rm emp}$, which were addressed in the rebuttal. One unresolved issue was the relative lack of novelty: while the analysis does require new ideas, it is somewhat restricted in scope ($\ell^p$ norms with $1\leq p\leq 2$) and strongly related to reference [BGM23]. My own evaluation is that the paper does present some interesting ideas, but with certain limitations that make it not too appealing to the NeurIPS audience.